# TRPM channels mediate learned pathogen avoidance following intestinal distention

Adam Filipowicz, Jonathan Lalsiamthara, Alejandro Aballay*

Department of Molecular Microbiology & Immunology, Oregon Health & Science University, Portland, United States

**Abstract** Upon exposure to harmful microorganisms, hosts engage in protective molecular and behavioral immune responses, both of which are ultimately regulated by the nervous system. Using the nematode *Caenorhabditis elegans*, we show that ingestion of *Enterococcus faecalis* leads to a fast pathogen avoidance behavior that results in aversive learning. We have identified multiple sensory mechanisms involved in the regulation of avoidance of *E. faecalis*. The G-protein coupled receptor NPR-1-dependent oxygen-sensing pathway opposes this avoidance behavior, while an ASE neuron-dependent pathway and an AWB and AWC neuron-dependent pathway are directly required for avoidance. Colonization of the anterior part of the intestine by *E. faecalis* leads to AWB and AWC mediated olfactory aversive learning. Finally, two transient receptor potential melastatin (TRPM) channels, GON-2 and GTL-2, mediate this newly described rapid pathogen avoidance. These results suggest a mechanism by which TRPM channels may sense the intestinal distension caused by bacterial colonization to elicit pathogen avoidance and aversive learning by detecting changes in host physiology.

## Introduction

In order to survive, animals have evolved mechanisms to detect and avoid harmful organisms, including pathogenic bacteria (*Kavaliers et al., 2019*; *Medzhitov et al., 2012*). Understanding this complex survival mechanism will ultimately take an interdisciplinary approach. Genetically tractable organisms such as *Caenorhabditis elegans* and *Drosophila melanogaster* have proven to be quite useful in uncovering the molecular and cellular components of pathogen avoidance. In response to pathogen exposure, these relatively simple animals are capable of eliciting avoidance behaviors (*Anderson and McMullan, 2018*; *Cao et al., 2017*; *Meisel and Kim, 2014*; *Meisel et al., 2014*; *Pradel et al., 2007*; *Reddy et al., 2009*; *Singh and Aballay, 2019a*; *Singh and Aballay, 2019b*; *Soldano et al., 2016*; *Stensmyr et al., 2012*; *Styer et al., 2008*; *Tran et al., 2017*; *Zhang et al., 2005*), along with the activation of well-described microbicidal innate immune pathways (*Engelmann et al., 2011*; *Irazoqui et al., 2010*; *Singh and Aballay, 2019b*; *Styer et al., 2008*), both of which are regulated by the nervous system (*Singh and Aballay, 2020*; *Wani et al., 2020*).

It was recently proposed that bacterial colonization causes a distension of the *C. elegans* intestine that may be used as a general mechanism to activate immune pathways and pathogen avoidance (*Kumar et al., 2019*; *Singh and Aballay, 2019a*; *Singh and Aballay, 2019b*). However, how intestinal distension leads to avoidance remains unclear. One possibility is that mechanoreceptors on intestinal cells sense the pressure caused by distension and then send neuropeptide signals to the nervous system to initiate avoidance. While no mechanoreceptors have been found to be functional in the *C. elegans* intestine, it does express transient receptor (TRP) channels that could act as mechanoreceptors (*Xiao and Xu, 2011*). However, TRP channels are known to play many other roles, including but not limited to taste, thermoregulation, ion homeostasis, and pacemaker activity (*Holzer, 2011*). Indeed, while the TRP vanilloid (TRPV) channel genes *ocr-2* and *osm-9* are known to inhibit avoidance of the pathogen *Pseudomonas aeruginosa* (*Singh and Aballay, 2019a*), this

*For correspondence:
aballay@ohsu.edu

Competing interests: The authors declare that no competing interests exist.

inhibition is through their role on hyperoxia avoidance rather than direct sensing of bacteria or distention. Moreover, they are not expressed in the *C. elegans* intestine. One TRP channel that is expressed in the *C. elegans* intestine is the thermosensitive channel TRPA-1 (*Xiao et al., 2013*). TRPA-1 also plays a role in sensory neuron-mediated nose-touch responses (*Kindt et al., 2007*). Interestingly, the *Drosophila* homolog of this channel was shown to be involved in avoidance of bacterial lipopolysaccharide, though activity of this channel was localized to gustatory neurons (*Soldano et al., 2016*). The mammalian digestive system expresses many TRP channels, including members of the TRPA, TRP melastatin (TRPM), and TRPV families (*Holzer, 2011*). Because nearly all of these channels are conserved across species, identification of their role in pathogen avoidance in *C. elegans* is likely to improve our understanding of pathogen avoidance in a range of organisms.

Here, we present evidence that *C. elegans* avoids *Enterococcus faecalis* faster than *P. aeruginosa*. In contrast to the *P. aeruginosa*-mediated avoidance that requires small RNAs (sRNAs) (*Kaletsky et al., 2020*), *E. faecalis*-mediated avoidance does not depend on sRNAs, nor does it require ASI neurons, full bacterial virulence, or innate immune activation. Instead, avoidance occurs after a rapid expansion of the anterior part of the intestine. This avoidance behavior, which exists in opposition to an NPR-1-mediated hyperoxia avoidance process, requires the ASE, AWB, and AWC chemosensory neurons. These latter two neuron pairs regulate an olfactory aversive learning process that underpins avoidance of *E. faecalis.* Finally, we identified two novel regulators of intestinal distention-induced pathogen avoidance: the TRP melastatin (TRPM) channels GON-2 and GTL-2.

## Results

### *E. faecalis* elicits fast avoidance in *C. elegans*

*C. elegans* is a free-living nematode that feeds on organic material rich in microorganisms, and therefore frequently encounters pathogenic bacteria. It has developed behavioral strategies to minimize exposure to these threats. For example, it avoids the pathogenic *P. aeruginosa* bacteria slowly, taking between 12 and 24 hr to execute aerotactic and olfactory aversive learning processes depending on the bacterial growth conditions used (*Singh and Aballay, 2019a*; *Zhang et al., 2005*). In contrast, we previously reported that animals quickly avoid the lawns of the Gram-positive pathogens *E. faecalis*, *E. faecium*, and *S. aureus* (*Singh and Aballay, 2019b*). Examining this in more detail, we found that animals displayed strong, fast avoidance of monoaxenic lawns of *E. faecalis*, *E. faecium*, or *S. aureus* grown on brain–heart infusion (BHI) media (*Figure 1A,B*, *Figure 1—figure supplement 1*). While population avoidance levels peaked at 4 hr for *E. faecalis*, wild-type animals first leave *E. faecalis* lawns at around 19 min on average (*Figure 1C*). Interestingly, we observed animals frequently exiting and entering the bacterial lawns over time. Quantification of these events revealed a slight imbalance favoring exiting events as early as 1 hr, with this difference growing larger in the 3–4 hr window due to a decrease in the entry events (*Figure 1D*).

The fast exiting events induced by *E. faecalis* are reminiscent of a *C. elegans* rapid aversive behavior elicited by exposure to a dry drop of dodecanoic acid, a toxin secreted by *Streptomyces* (*Tran et al., 2017*). However, implementing this dry-drop assay with *E. faecalis* or *E. faecium* did not induce an aversive response (*Figure 1—figure supplement 1*), indicating that direct contact with live replicating bacteria is required for the rapid avoidance of *E. faecalis* or *E. faecium*. Rapid aversion alone also cannot explain the re-entry events. Instead, it is likely that there is a balance of attraction and aversion to *E. faecalis* that changes over the course of exposure. Thus, the animals learn to avoid *E. faecalis* quickly after exposure. This process likely involves sensory pathways, as animals with loss-of-function mutations in *tax-2*, a subunit of a cGMP-gated-ion-channel important for sensory neuron function (*Coburn and Bargmann, 1996*), show slowed avoidance (*Figure 1C*). Because it has recently been suggested that the sensation of a bacterial sRNAs is at least partly responsible for learned *P. aeruginosa* avoidance (*Kaletsky et al., 2020*), we examined whether RNA from *E. faecalis* may also elicit avoidance. Unlike RNA from *P. aeruginosa*, RNA from *E. faecalis* fails to induce avoidance (*Figure 1E*). The fact that exposure to bacterial sRNAs only accounts for ~25% of the avoidance of *P. aeruginosa,* and that they have no role in avoidance of *E. faecalis*, suggests that another pathway is required to induce pathogen avoidance.

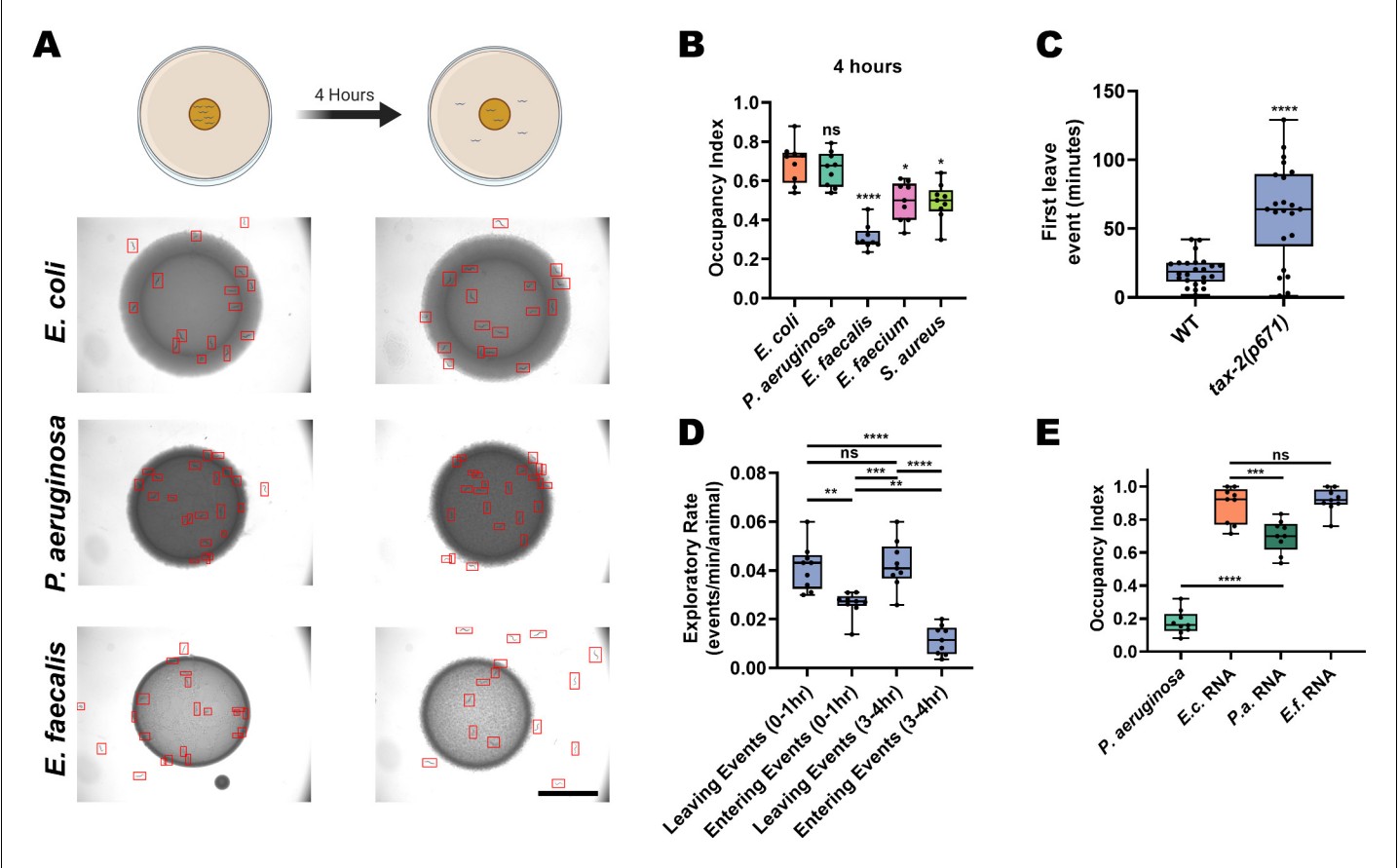

**Figure 1.** *E. faecalis* elicits fast avoidance in *C. elegans*. (A) Schematic of avoidance assays (top) and representative photomicrographs (bottom) of *C. elegans* on lawns of *E. coli* OP50, *P. aeruginosa* PA14, or *E. faecalis* OG1RF at 0 hr (left) and 4 hr (right) on BHI media. Individual animals are outlined in red. Scale bar, 5 mm. (B) Occupancy index of N2 animals after 4 hr of incubation on *E. coli*, *P. aeruginosa*, *E. faecalis*, *E. faecium*, or *S. aureus*. One-way ANOVA with subsequent comparison to *E. coli* as the control group was performed. Occupancy Index = (number of animals on bacterial lawn)/(total number of animals). (C) Individual animals were tracked on lawns of *E. faecalis* and the time that they first left the lawn after transfer was recorded. Wild-type (WT, N2 Bristol) animals showed an average first leave time of 19.20 min while *tax-2(p671)* animals showed an average first leave time of 61.98 min. An unpaired t-test between the groups was performed. (D) N2 animals on standard avoidance assays plates with *E. faecalis* were observed for 10 min during two different time windows (0–1 hr and 3–4 hr) and the number of times the animals left and entered the bacteria lawns was counted. One-way ANOVA with subsequent comparisons between all groups was performed. (E) Occupancy index of N2 animals after 24 hr incubation on *P. aeruginosa* PA14 lawns or *E. coli* OP50 lawns supplemented with isolated total RNA from *E. coli* (*E.c.* RNA), *P. aeruginosa* (*P.a.* RNA), or *E. faecalis* (*E.f.* RNA). One-way ANOVA with subsequent comparison to *E. coli* RNA as the control group was performed.

The online version of this article includes the following source data and figure supplement(s) for figure 1:

**Figure supplement 1.** Avoidance of Gram-positive pathogens.

**Figure supplement 1—source data 1.** Raw data for *Figure 1—figure supplement 1A*.

## Avoidance of *E. faecalis* follows anterior intestinal distention and is independent of virulence

Two other potential pathways for induction of avoidance are sensation of intestinal distention or bacterial metabolites. The bacterial metabolite serrawettin W2 is responsible for the avoidance of *S. marcescens* (*Pradel et al., 2007*). Even though secondary metabolites phenazine-1-carboxamide and pyochelin produced by *P. aeruginosa* are insufficient for the elicitation of pathogen avoidance (*Singh and Aballay, 2019a*), they are sensed by *C. elegans* as they elicit induction of DAF-7/TGF-β in ASJ neurons (*Meisel et al., 2014*). However, *E. faecalis* does not increase expression of DAF-7 in the ASJ neurons (*Meisel et al., 2014*), further strengthening the idea that direct contact with this pathogen rather than metabolite sensing induces avoidance.

To determine whether intestinal distention plays a role in avoidance of *E. faecalis*, we measured the diameter of the intestinal lumen of animals fed either *E. coli*, *P. aeruginosa*, or *E. faecalis* grown on BHI. There was significant expansion of the anterior part of the intestine of animals feeding on *E. faecalis* compared to *E. coli* even at 1 hr, with distention persisting at 4 hr (*Figure 2A,B*). The medial part of the intestine was not significantly distended at 1 hr compared to animals fed *E. coli*, but was distended at 2 and 4 hr. The anterior part of the intestine was slightly distended in animals feeding on *P. aeruginosa* compared to *E. coli* at 4 hr, but was otherwise not distended. Both parts of the intestine, but especially the anterior portion, were filled with live *E. faecalis*, as determined through the use of an *E. faecalis* strain expressing GFP, while the intestine of animals fed *E. coli* or *P. aeruginosa* expressing GFP were relatively empty and not distended (*Figure 2C*). Because we first measured the anterior intestinal diameter at 1 hr, it was not clear whether distention preceded

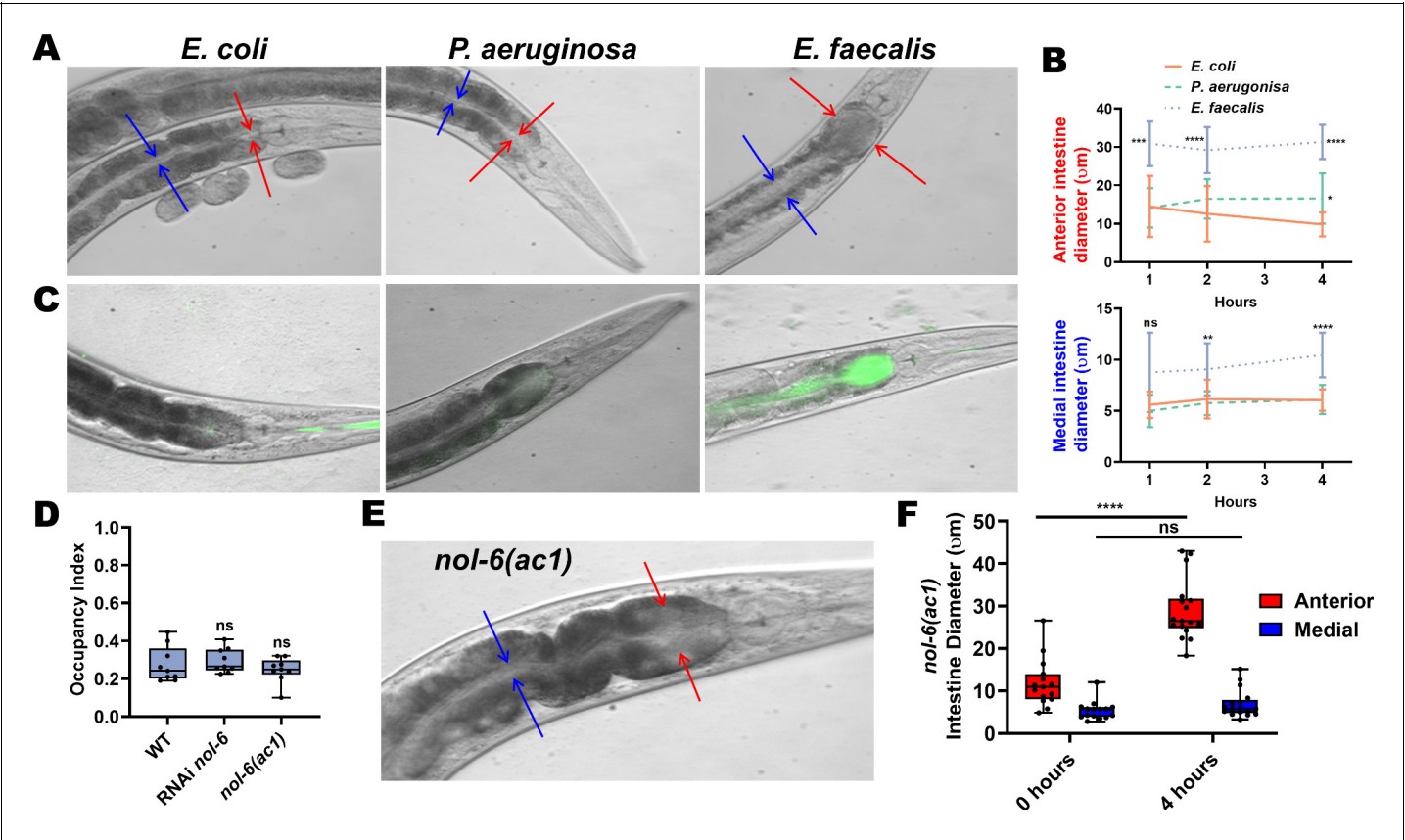

**Figure 2.** Anterior intestinal distention elicits avoidance of *E. faecalis*. (**A**) Representative photomicrographs of animals at 4 hr on BHI media with *E. coli* (left), *P. aeruginosa* (middle), or *E. faecalis* (right). Red arrows point to the borders of the anterior intestine while blue arrows point the borders of the medial intestine. (**B**) Quantification of the anterior (top) and medial (bottom) intestinal lumen diameter of animals on *E. coli*, *P. aeruginosa*, or *E. faecalis* at 1, 2, and 4 hr. Two-way ANOVA was performed with comparison to the *E. coli* group at each time point. For *E. coli*, N = 11, 13, and 17 at 1, 2, and 4 hr, respectively. For *P. aeruginosa*, N = 18, 14, and 18. For *E. faecalis*, N = 13, 18, and 16. Points are the mean of each group and error bars are standard deviation. (**C**) Representative photomicrographs with fluorescent and brightfield images merged of *E. coli*, *P. aeruginosa*, and *E. faecalis* expressing GFP in the intestinal lumen of animals at 4 hr. (**D**) Occupancy index on *E. faecalis* at 4 hr of WT animals compared to either RNAi-mediated knockdown of *nol-6* or a loss-of-function mutation in *nol-6*. One-way ANOVA was performed with subsequent comparison to the WT group as the control. (**E**) A representative photomicrograph of a *nol-6(ac1)* animal after exposure to *E. faecalis* for 4 hr. Red arrows point to the borders of the anterior intestine, and blue arrows point to the borders of the medial intestine. (**F**) Quantification of anterior and medial intestinal lumen diameter of *nol-6(ac1)* animals at 0 and 4 hr on *E. faecalis*. Two-way ANOVA was performed with comparison to the 0 hr groups for both anterior and medial quantifications.

The online version of this article includes the following source data and figure supplement(s) for figure 2:

**Source data 1.** Raw data for *Figure 2B*.

**Figure supplement 1.** Animals evacuating *E. faecalis* lawns show significant anterior intestinal distention.

**Figure supplement 2.** *E. faecalis* but not *P. aeruginosa* causes anterior intestinal distention in *nol-6* animals.

avoidance or vice versa. To explore this, we tracked animals until they left a lawn of *E. faecalis* and measured their anterior intestines (*Figure 2—figure supplement 1*). These animals had significantly larger anterior intestines compared to animals that remained on the *E. faecalis* lawn. It is therefore likely that fast accumulation of *E. faecalis* in the anterior intestine induces an aversive response to the pathogen, leading to avoidance.

RNA interference (RNAi)-mediated knockdown of *nol-6*, a nucleolar RNA-associated protein, reduces colonization and distention of the intestine by microbial pathogens (*Fuhrman et al., 2009*). Furthermore, knockdown of *nol-6* delays avoidance of *P. aeruginosa* by delaying intestinal distention (*Singh and Aballay, 2019a*; *Figure 2—figure supplement 2*). Therefore, we tested both *nol-6* RNAi and a *nol-6* loss-of-function (lf) mutant (*nol-6(ac1)*) for avoidance of *E. faecalis*. Avoidance for both the mutant and *nol-6* RNAi remained at wild-type levels (*Figure 2D*). Interestingly, while anterior intestinal distension was observed in both *nol-6(ac1)* and *nol-6* RNAi animals fed *E. faecalis* at 4 hr, the middle part of the intestine did not display significant distention (*Figure 2E,F*, *Figure 2—figure supplement 2*). This suggests that the anterior, but not medial, intestinal distention triggers avoidance of *E. faecalis*.

To determine whether bacterial virulence is required for anterior intestine distention-induced avoidance of *E. faecalis*, we used a strain of *E. faecalis* lacking the *fsrB* gene (*E. faecalis ΔfsrB*), a gene important for quorum sensing, which displays significantly reduced virulence across multiple animal models (*Garsin et al., 2001*; *Mylonakis et al., 2002*; *Sifri et al., 2002*). Surprisingly, this avirulent strain also elicits avoidance (*Figure 3A*) and causes anterior intestinal distention (*Figure 3B*). Lending further evidence to the notion that virulence plays no role in avoidance, *E. faecium*, a related enterococcal species that is non-pathogenic to *C. elegans* (*Garsin et al., 2001*; *Yuen and Ausubel, 2018*), also elicits avoidance (*Figure 1B*). While *E. faecium* is non-pathogenic to *C. elegans*, it does cause an innate immune response (*Yuen and Ausubel, 2018*), raising the possibility that an early immune response may be responsible for alerting the animal to avoid *E. faecalis* and related species upon ingestion.

Therefore, we examined the in vivo expression of two immune pathway markers, *clec-60* (*Irazoqui et al., 2010*; *Yuen and Ausubel, 2018*) and *ilys-3* (*Gravato-Nobre et al., 2016*; *Irazoqui et al., 2010*; *Yuen and Ausubel, 2018*), to determine whether an immune response was mounted at these early time points. While a later immune response could be observed, there was no evidence of an immune response at time points where early avoidance of *E. faecalis* is observed (*Figure 3C–E*). Furthermore, loss-of-function mutants for three key immune signaling genes, *pmk-1*, *fshr-1*, and *bar-1*, displayed wild-type levels of avoidance of *E. faecalis* (*Figure 3—figure supplement 1*). Altogether, these results suggest that virulence and immune pathway activation is not necessary for anterior distention and avoidance of *E. faecalis*, differentiating this avoidance mechanism further from that induced by *P. aeruginosa*, which requires virulence and intestine-wide distention (*Kumar et al., 2019*; *Singh and Aballay, 2019a*; *Singh and Aballay, 2019b*).

## TAX-2/4 pathways regulate avoidance of *E. faecalis* and *P. aeruginosa*

NPR-1, known to play a role in *C. elegans* survival and behavior on pathogens (*Meisel and Kim, 2014*; *Reddy et al., 2009*; *Styer et al., 2008*), is responsible for the intestinal distention-induced avoidance behavior on *P. aeruginosa*. We hypothesized that NPR-1 would also play a role in avoidance of *E. faecalis*. As a first step to test this, we measured avoidance of *E. faecalis* in two *npr-1 (lf)* mutants and found that both displayed decreased avoidance compared to wild-type animals (*Figure 4A*).

Inhibition of NPR-1 elicits avoidance of high oxygen (*de Bono and Bargmann, 1998*; *Chang et al., 2006*; *Rogers et al., 2003*), resulting in suppression of *P. aeruginosa* avoidance (*Reddy et al., 2009*; *Styer et al., 2008*), as the bacterial lawns have low oxygen due to microbial metabolism (*Reddy et al., 2011*). To test whether that is the case with avoidance of *E. faecalis*, wild-type and *npr-1 (lf)* animals were placed into hypoxia (8% oxygen) chambers and scored for avoidance. In this low oxygen environment, both *npr-1(ad609)* and *npr-1(ok1447)* displayed wild-type levels of avoidance (*Figure 4B*). Furthermore, when animals were taken out of the low oxygen environment and allowed to roam at atmospheric oxygen levels for 1-hr, wild-type animals remained off the *E. faecalis* lawns, while both *npr-1(lf)* strains migrated back onto the lawns (*Figure 4C*), indicating that hyperoxia avoidance is a more potent aversive stimulus than *E. faecalis* in *npr-1(lf)* animals.

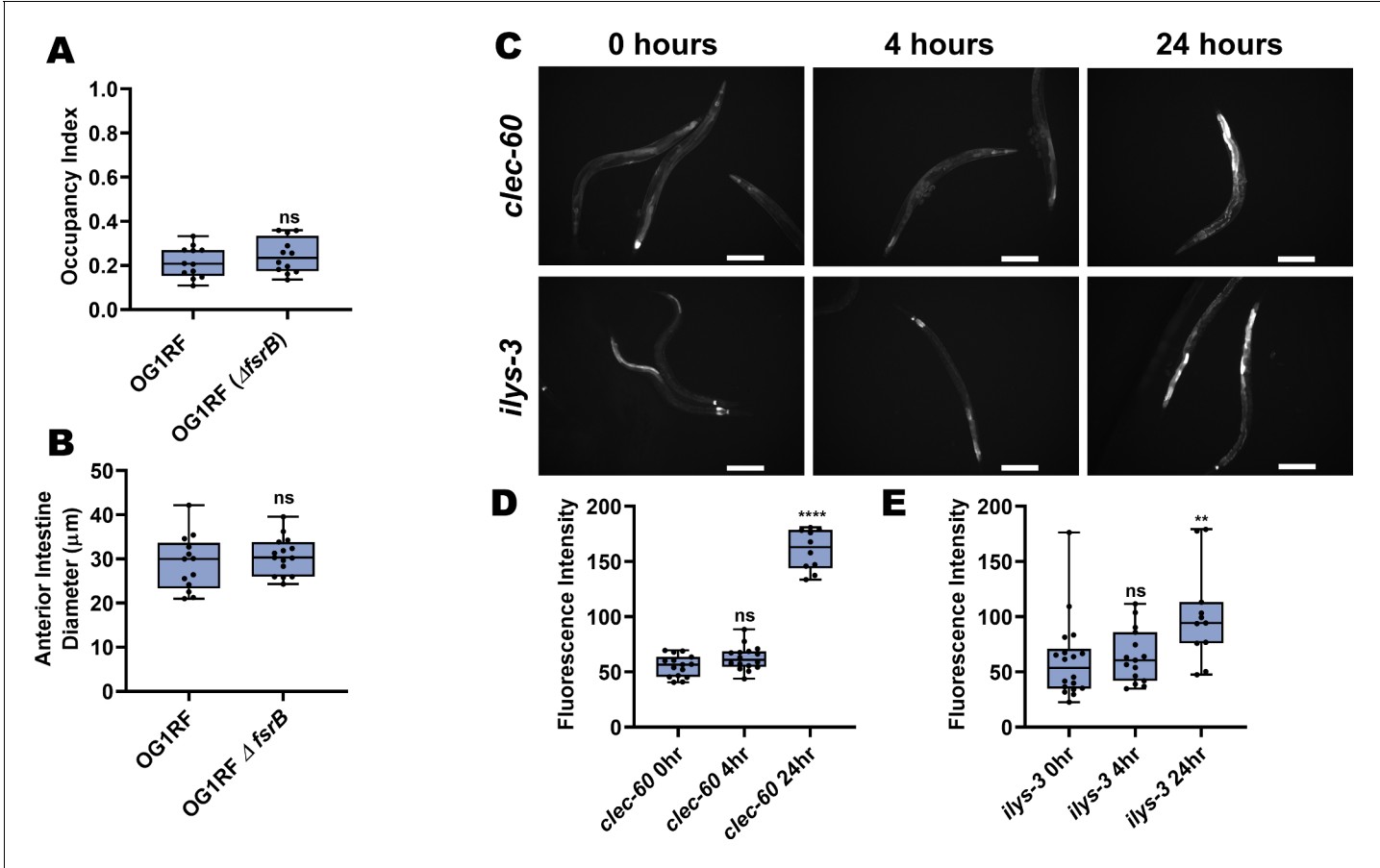

**Figure 3.** Avoidance of *E. faecalis* is independent of virulence. (**A**) Occupancy index of N2 animals on virulent *E. faecalis* OG1RF and avirulent *E. faecalis* OG1RF Δ*fsrB* at 4 hr. An unpaired t-test was performed. (**B**) Quantification of anterior intestinal diameter of N2 animals on *E. faecalis* strain OG1RF or OG1RF Δ*fsrB* at 4 hr. An unpaired t-test was performed. (**C**) Representative fluorescent micrographs of *clec-60p::GFP* (top) and *ilys-3p::GFP* (bottom) animals at 0, 4, and 24 hr on *E. faecalis*. Scale bar, 200 μm. (**D**) Quantification of *clec-60::GFP* at 0, 4, and 24 hr on *E. faecalis*. One-way ANOVA was performed with comparisons to the 0 hr group as the control. (**E**) Quantification of *ilys-3p::GFP* at 0, 4, and 24 hr on *E. faecalis*. One-way ANOVA was performed with comparisons to the 0 hr group as the control.

The online version of this article includes the following figure supplement(s) for figure 3:

**Figure supplement 1.** Loss of key immune genes does not affect avoidance of *E. faecalis*.

Hyperoxia avoidance in *npr-1(lf)* animals requires the transient receptor potential channel vanilloid (TRPV) genes *ocr-2* and *osm-9* (*de Bono et al., 2002*; *Chang et al., 2006*; *Singh and Aballay, 2019a*). Loss-of-function mutations in these genes results in decreased hyperoxia avoidance, and should thus lead to increased pathogen avoidance. Indeed, while loss-of-function mutants in both genes displayed avoidance levels similar to wild-type animals at 4 hr, their rate of avoidance was faster than that of wild-type animals (*Figure 4—figure supplement 1*). Hyperoxia avoidance of *npr-1(lf)* nematodes also depends on functional *gcy-35*, *tax-2*, and *tax-4* genes (*Chang et al., 2006*; *Gray et al., 2004*). GCY-35 is a soluble guanylyl cyclase that binds directly to molecular oxygen (*Cheung et al., 2004*), while TAX-2 and TAX-4 are two subunits of a cGMP-gated-ion-channel (*Coburn and Bargmann, 1996*) thought to act downstream of GCY-35 (*Coates and de Bono, 2002*; *Gray et al., 2004*). Through the activity of GCY-35 and TAX-2/TAX-4, the sensory neurons AQR, PQR, and URX drive avoidance of high oxygen. We tested double loss-of-function mutants of *gcy-35*, *tax-2*, or *tax-4* and *npr-1*, hypothesizing that *gcy-35*, *tax-2*, and *tax-4* mutations would suppress the lack of pathogen avoidance of *npr-1(ad609)* animals. While this was indeed the case for the *gcy-35* mutation, *tax-2* and *tax-4* mutations failed to suppress the lack of avoidance of *E. faecalis* exhibited by *npr-1(ad609)* animals (*Figure 4D*). These results suggest that TAX-2 and TAX-4 are involved in the detection of avoidance cues in addition to their role in oxygen detection.

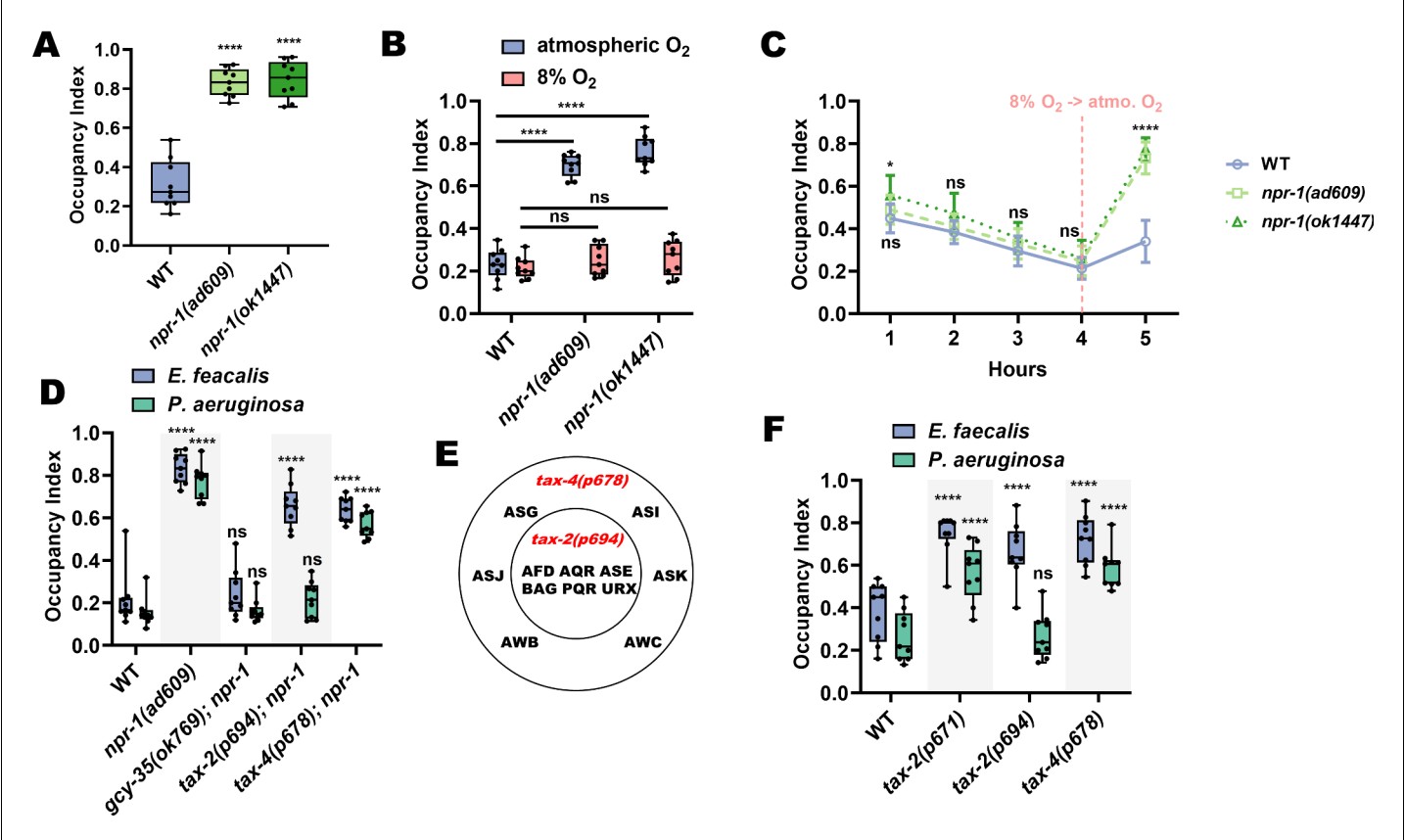

**Figure 4.** TAX-2/4 pathways regulate avoidance of *E. faecalis* and *P. aeruginosa*. (A) Occupancy index of WT animals compared to *npr-1(ad609)* and *npr-1(ok1447)* after 4 hr on *E. faecalis.* One-way ANOVA with subsequent comparison to WT animals as the control group was performed. (B) Occupancy index of WT, *npr-1(ad609)* and *npr-1(ok1447)* on lawns of *E. faecalis* in atmospheric oxygen or a chamber containing 8% oxygen. Two-way ANOVA with subsequent comparison to WT animals of each respective oxygen condition was performed. (C) Occupancy index of WT, *npr-1(ad609)*, and *npr-1(ok1447)* in an 8% oxygen chamber at 1, 2, 3, and 4 hr, and 1 hr after removal from the chamber (vertical dashed line). Two-way ANOVA with subsequent comparison at each time point to WT animals was performed. N = 9 for all animals. Points are the mean of each group and error bars are standard deviation. (D) Occupancy index for WT and double loss-of-function mutants for *gcy-35*, *tax-2*, or *tax-4*, and *npr-1* on *E. faecalis* for 4 hr and *P. aeruginosa* for 24 hr. Two-way ANOVA with subsequent comparisons to WT animals for each bacterium were performed. (E) Diagram of the neurons affected by *tax-2(p694)* and *tax-4(p678)*. The latter allele covers all *tax-2* expressing sensory neurons, while the former covers a subset. (F) Occupancy index for WT and single loss-of-function mutants for *tax-2* or *tax-4* on *E. faecalis* for 4 hr and *P. aeruginosa* for 24 hr. Two-way ANOVA as in with subsequent comparisons to WT animals for each bacterium were performed.

The online version of this article includes the following source data and figure supplement(s) for figure 4:

**Source data 1.** Raw data for *Figure 4C*.

**Figure supplement 1.** The TRPV subunits OCR-2 and OSM-9 slightly speed up avoidance of *E. faecalis*.

**Figure supplement 1—source data 1.** Raw data for *Figure 4—figure supplement 1B*.

We next tested the role of TAX-2 and TAX-4 in the NPR-1-mediated avoidance of *P. aeruginosa* using double mutants for *tax-2* or *tax-4* and *npr-1*. Surprisingly, and in contrast to avoidance of *E. faecalis*, *tax-2* but not *tax-4* mutation suppressed the avoidance of *P. aeruginosa* of *npr-1(ad609)* animals (*Figure 4D*). The discrepancy between the double mutants is that the *tax-2(p694)* allele used only affects a subset of the *tax-2* expressing neurons (*Harris et al., 2014*). The *tax-4(p678)* allele, in contrast, affects all *tax-4*-expressing neurons (*Figure 4E*). Thus, the neurons affected by the *tax-4 (p678)* allele, but spared by the *tax-2(p694)* allele, most likely play a role in avoidance of *P. aeruginosa* independent from the oxygen-sensing pathway. The results from the *E. faecalis* experiments, on the other hand, suggest that the *tax-2(p694)* allele covers additional neurons that regulate avoidance of that bacteria. To confirm this, we used single loss-of-function mutants for *tax-2* and *tax-4*, including a strain carrying an allele that affects expression in all *tax-2* expressing neurons (*tax-2*

(p671)). We indeed found that all strains tested displayed decreased avoidance of *E. faecalis* compared to wild-type animals, while only *tax-2(p671)* and *tax-4(p678)* mutants displayed decreased avoidance of *P. aeruginosa* (*Figure 4F*).

## ASE, AWB, and AWC neurons mediate avoidance of *E. faecalis*

We turned our attention to uncovering the specific neurons in the TAX-2/TAX-4-dependent pathway that regulate avoidance of both *E. faecalis* and *P. aeruginosa*. To identify the neuron(s) responsible for *E. faecalis* avoidance, we employed two strategies. First, we asked whether *tax-2* expression in individual neurons would rescue the lack of pathogen avoidance of *tax-2(p694)* animals. This revealed that *tax-2* expression in ASE neurons was sufficient for pathogen avoidance, while *tax-2* expression in the AQR, PQR, and URX neurons had no effect (*Figure 5A*). Second, we used animals lacking ASE neurons and the individual AWB, AWC, and ASI neurons, which are spared by the *tax-2 (p694)* allele. The results confirmed the involvement of ASE neurons in *E. faecalis* avoidance and showed that AWB and AWC neurons are also necessary (*Figure 5B*).

For avoidance of *P. aeruginosa*, we also employed the genetic ablation strategy, which revealed the requirement of AWB and AWC neurons (*Figure 5B*). Confirming the results of the *tax-2* experiments, ASE neurons were not required for avoidance of *P. aeruginosa*. We also confirmed a previous report that ASI neurons, but not ASH neurons, are required for avoidance of *P. aeruginosa* (*Cao et al., 2017*), while showing that neither is required for avoidance of *E. faecalis* (*Figure 5B*). Using a two-choice assay where, unlike the pathogen avoidance assay that only uses *P. aeruginosa*, animals are given a choice between *E. coli* and *P. aeruginosa*, it was shown that ASI neurons respond to *P. aeruginosa* RNAs by eliciting avoidance (*Kaletsky et al., 2020*). Using both avoidance and choice assays, we found that animals lacking ASI neurons fail to avoid *P. aeruginosa* after total RNA exposure (*Figure 5C,D*). These results are consistent with the idea that ASI neurons are capable of sensing *P. aeruginosa* RNAs and that *E. faecalis* RNAs do not induce avoidance (*Figure 1E*). Altogether, these results suggest that regulation of behavioral immunity is pathogen specific (*Figure 5E*).

## AWB and AWC neurons are necessary for aversive olfactory learning following ingestion of *E. faecalis*

Our data suggested that avoidance of *E. faecalis* represented an aversive learning process (*Figure 1D*), but we tested this directly. AWB and AWC neurons are known to play a role in odor preference and olfactory aversive learning in *C. elegans* in the context of *P. aeruginosa* (*Ha et al., 2010*; *Harris et al., 2014*). Because AWB and AWC neurons are also involved in avoidance of *E. faecalis*, we hypothesized that odor preference and olfactory aversive learning may also play a role in avoidance of *E. faecalis*. We first set out to establish the naïve preference of *C. elegans* when given the choice between *E. coli* and *E. faecalis*. We used a two-choice assay in which animals were able to roam for 1 hr on test plates containing *E. coli* and *E. faecalis* lawns on opposite sides of the plates before scoring animals based on lawn occupancy. Under this condition, animals showed a preference for *E. coli* over *E. faecalis*; however, if animals were instead paralyzed upon approaching the lawns, they displayed a preference for *E. faecalis* (*Figure 6A*). We hypothesized that there is an initial attraction to *E. faecalis* that is quickly overcome after feeding on it, such that animals go toward the familiar *E. coli* lawn instead. Paralyzing animals on lawn arrival captures the initial choice, while allowing them to freely roam captures the aversive learning behavior. We further hypothesized that this attraction-aversion dynamic was at least partly olfactory, based on neuron ablation results (*Figure 5*), and evidence that *tax-2(lf)* leads to significantly delayed lawn exiting (*Figure 1C*).

To directly test the idea of an initial odor attraction, we modified the choice assay by attaching plugs of agar with bacterial lawns on opposite sides of the lids of empty test plates (*Figure 6B*, schematic), preventing the animals from getting direct contact with the bacteria. We then placed the anesthetic sodium azide on the test plate surface underneath the bacteria plugs and placed animals in the center of the plate, allowing them to migrate toward one side or the other. This revealed a naïve preference for the odor of *E. faecalis* over *E. coli* (*Figure 6B*, left). Both *tax-2(p671)* and *tax-4 (p678)* mutants displayed decreased naïve odor preferences but *tax-2(p694)* did not. This result is consistent with the idea that the *tax-2* and *tax-4* expressing neurons AWB and AWC are responsible for establishing the naïve odor preference. Indeed, ablating them also resulted in a decreased

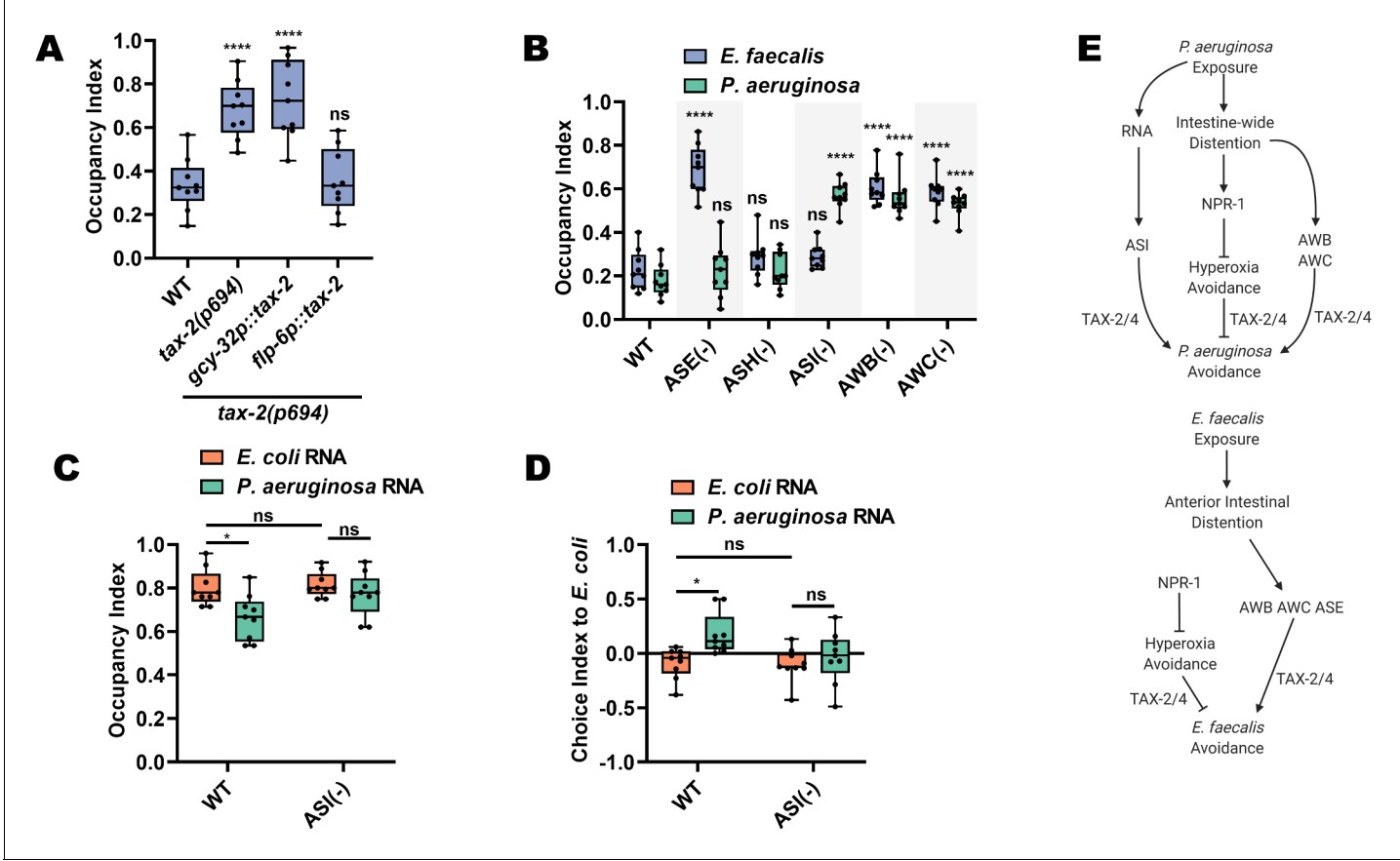

**Figure 5.** ASE, AWB, and AWC neurons mediate avoidance of *E. faecalis*. (**A**) Occupancy index on *E. faecalis* at 4 hr for WT, *tax-2(p694)* and animals with *tax-2* expression in ASE neurons (*flp-6p::tax-2*) or AQR, PQR, or URX neurons (*gcy-32p::tax-2*) in the *tax-2(p694)* background. One-way ANOVA with subsequent comparison to wild-type (WT) animals was performed. (**B**) Occupancy index of WT animals and animals with ablated neurons on *E. faecalis* at 4 hr and *P. aeruginosa* at 24 hr. Ablation of sensory neurons either by mutation (*che-1(p680)* = ASE(−)) or by caspase expression (*sra-6p::mCasp-1*=ASH(−); *gpa-4p::TU#813 + gcy-27p::TU#814* = ASI(−); *str-1p::mCasp-1* = AWB(−); *ceh-36p::TU#813 + ceh-36p::TU#814* = AWC(−)). Two-way ANOVA with subsequent comparisons to WT groups for each respective bacterium were performed. (**C**) Occupancy index at 1 hr for WT and ASI(−) animals on *P. aeruginosa* lawns. Animals were trained on *E. coli* OP50 lawns supplemented with either *E. coli* or *P. aeruginosa* RNA for 24 hr. Two-way ANOVA with subsequent comparison with WT control groups was performed. (**D**) Choice index after 1 hr for WT and ASI(−) animals choosing between *E. coli* and *P. aeruginosa* lawns. Two-way ANOVA with subsequent comparison with WT control groups was performed. Choice Index = (number of animals on *E. coli* − number of animals on *P. aeruginosa*)/(number of animals on *E. coli* + number of animals on *P. aeruginosa*). (**E**) Model for avoidance of *P. aeruginosa* (top) and *E. faecalis* (bottom) Avoidance of *P. aeruginosa* depends on both an ASI neuron-mediated bacterial sRNA pathway along with intestine-wide distention. The latter requires the NPR-1-dependent hyperoxia avoidance pathway along with AWB and AWC olfactory neurons. Avoidance of *E. faecalis* also depends on intestinal distention, though this is confined to the anterior intestine. This anterior intestinal distention-induced avoidance also requires AWB and AWC olfactory neurons, but also requires ASE chemosensory neurons. NPR-1-dependent hyperoxia avoidance opposes *E. faecalis* avoidance and depends on functional TAX-2/4.

preference (*Figure 6B*, left). We next asked whether trained animals that were allowed to ingest *E. faecalis* would switch their odor preference. The animals were trained by exposing them to *E. faecalis* for 4 hr using the same plate setup that had been used for the avoidance assays. Then, the trained animals were transferred to the lid-choice testing plates. A marked shift in preference was observed for trained wild-type animals that was not seen in *tax-2(p671)*, *tax-4(p678)*, or AWB- and AWC-ablated strains (*Figure 6B*, left). The *tax-2(p694)* mutants, on the other hand, behaved like wild-type animals. The learning index for each strain also highlights that aversive olfactory learning takes place in wild-type and *tax-2(p694)* animals but not in *tax-2(p671)* or *tax-4(p678)* mutants or AWB- and AWC-ablated strains (*Figure 6B*, right). Thus, the *tax-2* and *tax-4* expressing AWB and AWC neurons are necessary both for establishing a naïve preference and for aversive olfactory learning following ingestion of *E. faecalis* over a period of 4 hr.

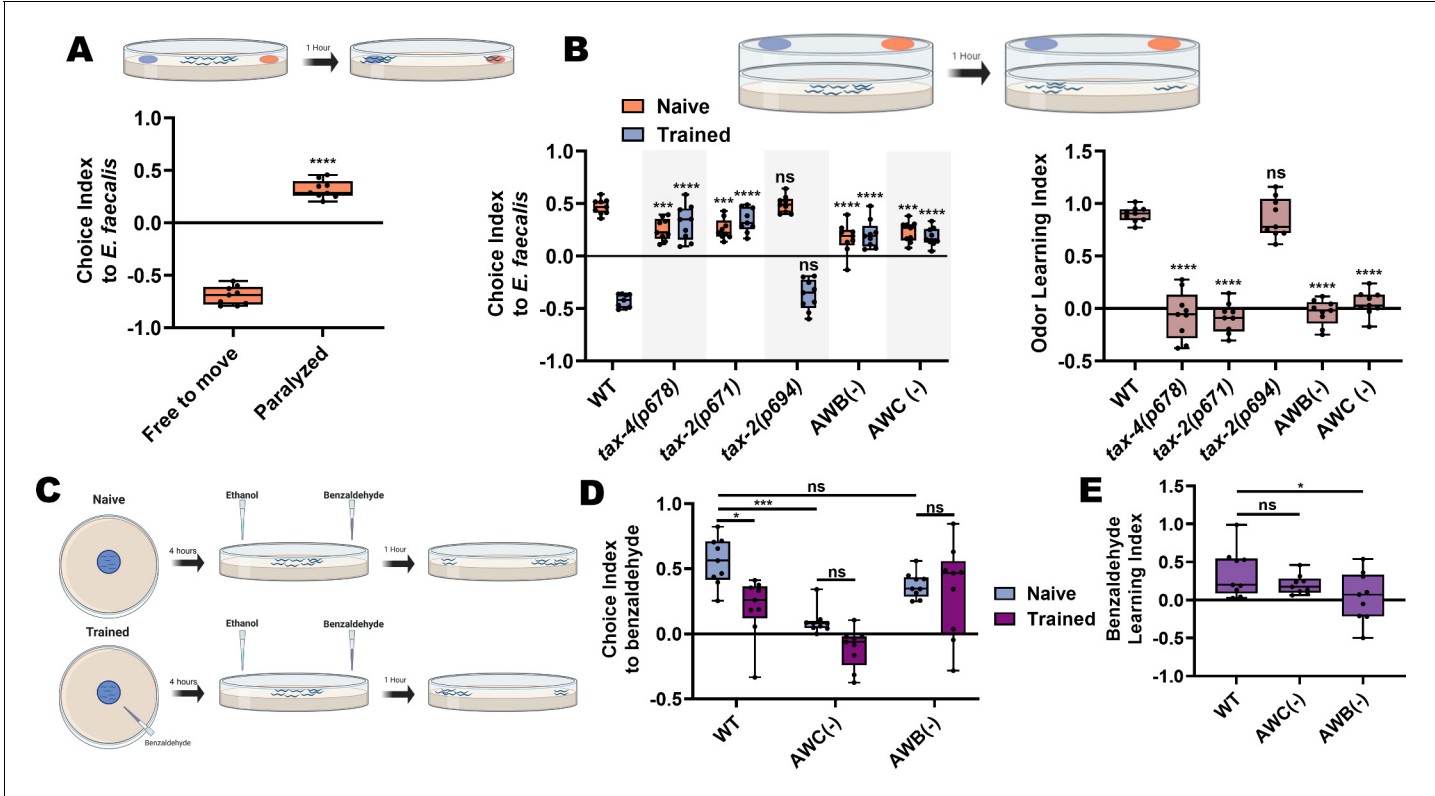

**Figure 6.** AWB and AWC neurons are necessary for aversive olfactory learning following ingestion of *E. faecalis*. (**A**) Schematic of the choice assay (top) and quantification (bottom) of choice index for N2 animals choosing between *E. coli* and *E. faecalis* lawns and either free to move for 1 hr (Free to move) or paralyzed upon arrival at a bacterial lawn (Paralyzed). An unpaired t-test between the groups was performed. (**B**) Schematic of the lid choice assay (top) and quantification of naïve and trained choice index (bottom, left) and learning index (bottom, right) for WT, *tax-4(p678)*, *tax-2(p671)*, *tax-2 (p694)*, AWB(−), and AWC(−) animals. To train animals, young adult animals were placed on *E. faecalis* lawns for 4 hr before the choice assay was performed. For choice index, two-way ANOVA with subsequent comparisons to the naïve and trained WT groups as controls were performed. For learning index, one-way ANOVA with comparison to the WT group as control was performed. (**C**) Schematic of the paired olfactory choice assay. (**D**) Quantification of choice index for naïve and trained WT, AWC(−), and AWB(−) animals, choosing between 1:200 benzaldehyde and ethanol. Two-way ANOVA with subsequent comparisons between naïve and trained groups and between WT and neuron ablated animals was performed. (**E**) Quantification of the learning index for WT, AWC(−), and AWB(−) animals from (**D**). One-way ANOVA with comparison to the WT group as control was performed. Learning index = (naïve choice index) – (trained choice index).

To strengthen the idea that avoidance of *E. faecalis* involves aversive learning, we paired an attractive odorant, benzaldehyde, with lawns of *E. faecalis* and exposed animals to this pairing for 4 hr (*Figure 6C*). Animals were then transferred to a choice plate, spotted with the attractive odorant on one side and ethanol on the other, and were allowed to roam for 1 hr before scoring. Results were compared to a naïve group that had no exposure to the odorant. Naïve animals displayed a strong attraction to benzaldehyde that was significantly reduced upon training (*Figure 6D,E*). As benzaldehyde is primarily sensed by AWC neurons, ablation of these cells led to a decrease in naïve attraction, with training leading to no additional decrease. Interestingly, animals with ablated AWB neurons displayed wild-type levels of naïve attraction but did not show a decrease in attraction upon training, indicating that AWB neurons may be required for this aversive learning process.

## The TRPM channels *gon-2* and *gtl-2* are required for distention-induced pathogen avoidance

It remains unclear how *C. elegans* senses intestinal distention in order to execute pathogen avoidance, either for *P. aeruginosa* or for *E. faecalis*. Because TRP channels could be mechanoreceptors (*Xiao and Xu, 2011*) capable of sensing the intestinal distention caused by microbial colonization, we screened animals with loss-of-function mutations in five TRP-encoding genes for avoidance of *E. faecalis* and found that two members of the TRPM subfamily, *gon-2* and *gtl-2*, negatively affected

avoidance of *E. faecalis* (*Figure 7—figure supplement 1*). Because *gon-2* mutants are known to have abnormal gonad development, and the germline has previously been shown to be involved in pathogen avoidance and immunity (*Kaletsky et al., 2020*; *Moore et al., 2019*; *TeKippe and Aballay, 2010*), we tested whether the effect of *gon-2(lf)* mutations on pathogen avoidance was due to perturbed gonad development. We took advantage of the fact that the Gon phenotype is temperature sensitive (high penetrance at 25°C and low penetrance at 15°C) and compared avoidance results for both *E. faecalis* and *P. aeruginosa* at both temperatures. For *P. aeruginosa*, *gon-2(lf)* animals raised at 25°C failed to avoid, while animals raised at 15°C avoided like wild-type animals

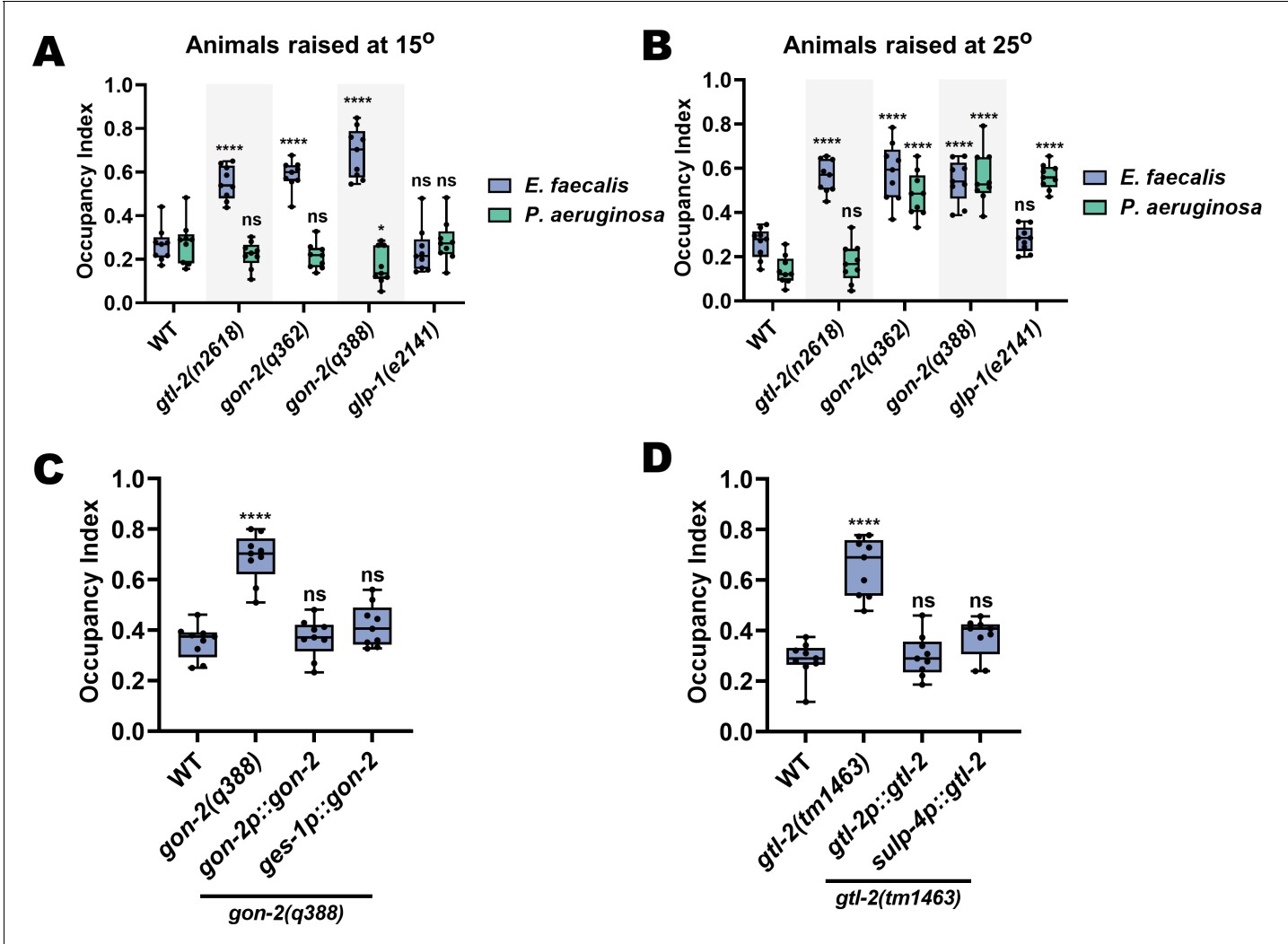

**Figure 7.** The TRPM channels GON-2 and GTL-2 are required for distention-induced pathogen avoidance. (A) Occupancy index for wild-type, *gtl-2 (n2618)*, *gon-2(q362)*, *gon-2(q388)*, and *glp-1(e2141)* animals on *E. faecalis* at 4 hr or *P. aeruginosa* at 24 hr. Animals were grown to the young adult stage at 15°C. Two-way ANOVA with comparisons to the respective WT group for each bacterium were performed. (B) Occupancy index for wild-type, *gtl-2(n2618)*, *gon-2(q362)*, *gon-2(q388)*, and *glp-1(e2141)* animals on *E. faecalis* at 4 hr or *P. aeruginosa* at 24 hr. Animals were grown to the young adult stage at 25°C. Two-way ANOVA with comparisons to the respective WT group for each bacterium were performed. (C) Occupancy index for wild-type, *gon-2(q388)*, self-promoter rescue (*gon-2(q388);gon-2p::gon-2*), and intestine specific rescue (*gon-2(q388);ges-1p::gon-2*) animals on *E. faecalis* at 4 hr. One-way ANOVA with comparison to the wild-type group as control was performed. (D) Occupancy index for wild-type, *gtl-2(tm1463)*, self-promoter rescue (*gtl-2(tm1463);gtl-2p::gtl-2*), and excretory cell specific rescue (*gtl-2(tm1463);sulp-4p::gtl-2*) animals on *E. faecalis* at 4 hr. One-way ANOVA with comparison to the wild-type group as control was performed.

The online version of this article includes the following figure supplement(s) for figure 7:

**Figure supplement 1.** Screening TRP channels for effect on avoidance of *E. faecalis*.

**Figure supplement 2.** Rescue of *gon-2* and *gtl-2* expression in mutant backgrounds.

(*Figure 7A,B*). For *E. faecalis*, *gon-2(lf)* animals failed to avoid at both temperatures (*Figure 7A,B*), suggesting that the germline does not play a role in the control of avoidance of this pathogen. Furthermore, *glp-1(e2141)* mutants, which lack most germline cells due to defects in mitotic and meiotic division (*Austin and Kimble, 1987*; *Crittenden et al., 1994*), behave similarly to *gon-2(lf)* mutant animals on both pathogens (*Figure 7A,B*). This suggests that *gon-2(lf)* animals fail to avoid *P. aeruginosa* specifically because of a lack of gonad development, while for *E. faecalis* the effect on avoidance is independent of the germline. Unlike *gon-2(lf)*, *gtl-2(n2618)* animals failed to avoid *E. faecalis*, but not *P. aeruginosa* (*Figure 7A,B*).

In addition to gonadal expression, *gon-2* is known to exhibit intestinal expression (*Teramoto et al., 2005*). To confirm this, we generated a strain expressing *gon-2::GFP* under the *gon-2* promoter, which revealed expression in the gonads and intestine (*Figure 7—figure supplement 2*). We therefore hypothesized that driving *gon-2* expression to the intestines of *gon-2(q388)* mutants would rescue the pathogen avoidance defect of the mutants. As shown in *Figure 7C*, *gon-2 (q388)* animals expressing *gon-2* under its own promoter or an intestine specific promoter (*ges-1p*) displayed wild-type levels of avoidance of *E. faecalis*. We also sought to uncover the site of action of *gtl-2* for the pathogen avoidance behavior. It was previously reported that *gtl-2* is functionally expressed in the excretory cell, with additional expression in the pharyngeal muscle (*Kwan et al., 2008*; *Teramoto et al., 2010*). We constructed a strain expressing *gtl-2::GFP* under its own promoter and found that this was indeed the case (*Figure 7—figure supplement 2*). Furthermore, *gtl-2 (tm1463)* mutants expressing *gtl-2* under its own promoter or under a promoter that drives expression only to the excretory cell (*sulp-4p*) displayed wild-type levels of avoidance (*Figure 7D*). Together, these results indicate that *gon-2* expression in the intestine and *gtl-2* expression in the excretory cell is sufficient for *C. elegans* avoidance of *E. faecalis*.

Because *gon-2(lf)* and *gtl-2(lf)* animals display wild-type levels of anterior intestinal distention (*Figure 8—figure supplement 1*), we hypothesized that the effect observed in these animals was due to an inability to sense intestinal distention either directly or indirectly. It was previously shown that knockdown of *aex-5* leads to defects in the defecation motor program and subsequent intestinal distention by the accumulation of bacteria, resulting in increased avoidance of even non-pathogenic *E. coli* (*Singh and Aballay, 2019a*). Interestingly, *gon-2(lf)* mutants failed to exhibit such avoidance on *E. coli* HT115 after knockdown of *aex-5*, while *gtl-2(n2618)* mutants behaved like wild-type animals (*Figure 8A*). These results are consistent with the different types of distention caused by *E. faecalis* and *P. aeruginosa*, with the former causing a more severe but localized distention of the anterior intestine and the latter causing a distention along the entire length of the intestine, which seems to require the germline to elicit avoidance. As shown in *Figure 8B*, both *gon-2* and *gtl-2* mutants suppressed the *aex-5* knockdown avoidance phenotype on *E. faecalis*, suggesting that both TRPM channels might be involved in the process of sensing the anterior intestinal distention caused by *E. faecalis*. However, TRPM channels display many distinct functions in various settings, including the absorption of calcium and magnesium ions by TRPM6 and TRPM7 in the human colon, TRPM7-mediated pacemaker activity of interstitial cells of Cajal, and colonic nociception by mouse TRPM8 (*Holzer, 2011*; *Mueller-Tribbensee et al., 2015*). Thus, it remains unclear how exactly GON-2 and GTL-2 function in pathogen avoidance.

If these TRPM channels are somewhat involved in sensing anterior intestinal distention caused by *E. faecalis* infection, their loss should leave naïve olfactory preferences to *E. faecalis* intact while diminishing negative associative learning. Testing *gon-2(lf)* and *gtl-2(lf)* mutants on olfactory choice assays with and without prior exposure to *E. faecalis* revealed that this was indeed the case (*Figure 8C,D*). All *gon-2* and *gtl-2* mutants displayed wild-type levels of naïve preference to *E. faecalis* versus *E. coli*. After training by exposure to *E. faecalis* for 4 hr, *gon-2(lf)* and *gtl-2(lf)* mutants displayed significantly reduced preference switching compared to wild-type animals (*Figure 8C*). This results in significantly lower learning indices for the mutants (*Figure 8D*). Altogether, these findings indicate that GON-2 and GTL-2 are required for the learned avoidance of *E. faecalis*, though their exact mechanistic role remains to be determined (*Figure 8E*).

## Discussion

This study establishes that *E. faecalis* infection in *C. elegans* leads to anterior intestinal distention which results in a rapid pathogen avoidance behavior, opposed by NPR-1-mediated hyperoxia

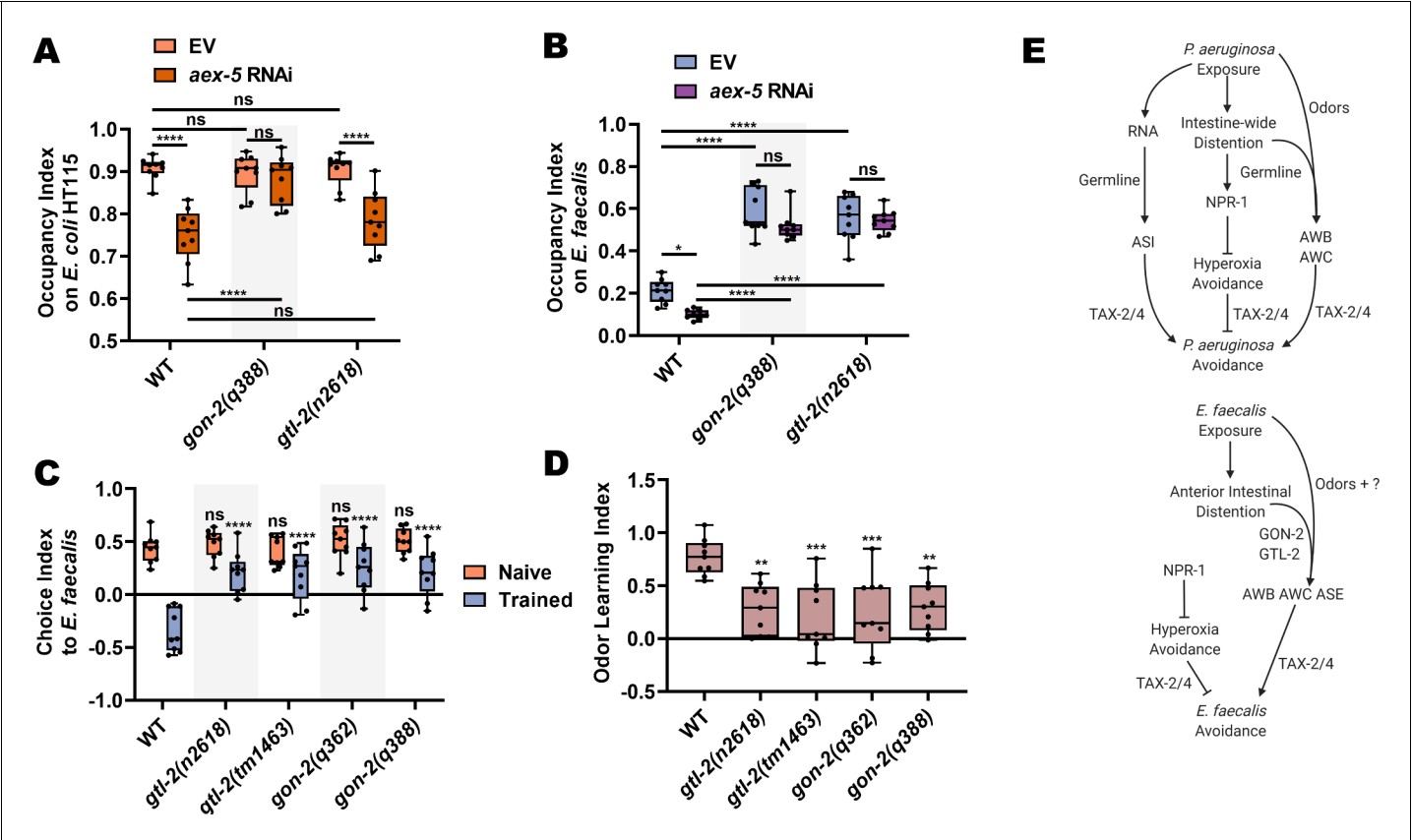

**Figure 8.** GON-2 and GTL-2 diminish olfactory aversive learning following *E. faecalis* exposure. (A) Occupancy index for WT, *gon-2(q388)*, and *gtl-2 (n2618)* animals with either RNAi-mediated knockdown of *aex-5* or an empty vector (EV) control on the *E. coli* HT115 lawns these animals were raised on. Two-way ANOVA with subsequent comparison between all groups was performed. (B) Occupancy index for WT, *gon-2(q388)*, and *gtl-2(n2618)* animals with either RNAi-mediated knockdown of *aex-5* or an EV control on *E. faecalis* lawns at 4 hr. Two-way ANOVA with subsequent comparison between all groups was performed. (C) Choice index for wild-type, *gtl-2(n2618)*, *gtl-2(tm1463)*, *gon-2(q362)*, and *gon-2(q388)* animals between *E. faecalis* and *E. coli* using the lid choice assay. Two-way ANOVA with comparison to the naïve and trained WT group was performed. (D) Quantification of learning index for animals from (C). One-way ANOVA with comparison to the WT group was performed. (E) Model for avoidance of *P. aeruginosa* (top) and *E. faecalis* (bottom), as in *Figure 5E*, with the addition of a germline role in avoidance of *P. aeruginosa*, GON-2 and GTL-2 regulation of avoidance of *E. faecalis*, and the contribution of odor sensing pathways to avoidance of both bacteria.

The online version of this article includes the following figure supplement(s) for figure 8:

**Figure supplement 1.** Loss of GON-2 or GTL-2 function does not affect anterior intestinal distention on *E. faecalis*.

avoidance, and regulated by TAX-2/4 expressing AWB, AWC, and ASE sensory neurons, and the TRPM channels GON-2 and GTL-2. In contrast, avoidance of *P. aeruginosa* uses different mechanisms that involve intestine-wide distention, germline mediated-signaling, and sensing of bacterial sRNAs via an ASI neuronal pathway (*Figure 8E*). Thus, bacterial context is important for the elicitation of pathogen avoidance behaviors. This context dependence makes it likely that using *C. elegans* may lead to the discovery of new components of sensory pathways involved in pathogen avoidance and help to elucidate how physiological information from the site of infection is relayed to the nervous system. This was indeed highlighted by the identification of ASE neurons and GTL-2 as being necessary for avoidance of *E. faecalis* but not *P. aeruginosa*, and the requirement of the germline in avoidance of *P. aeruginosa* but not *E. faecalis*.

The decision to leave a bacterial lawn represents a dynamic balance between risk and reward for *C. elegans*, with feeding and food choice being critical for survival and propagation (*Kiyama et al., 2012*; *Milward et al., 2011*; *Shtonda and Avery, 2006*). Initially, the balance may be toward staying on the lawn, as it could be an area of high reward in the form of food. However, depending on the quality of the food, its availability, and its potential noxious qualities, the balance may swing toward

bacterial avoidance, as the lawn becomes either an area of low reward (depleted or low-quality) or an area of high risk (pathogenic). The frequent entering and exiting events of animals on lawns of *E. faecalis* (*Figure 1C,D*) may be indicative of this dynamic balance. Multiple sensory inputs or physiological states such as oxygen level, odors, tastes, mechanosensation, pain, and hunger must be integrated in order to evaluate the environment and come to a decision. These cues are most likely integrated in the interneurons of *C. elegans*, which are downstream of the sensory neurons (*Metaxakis et al., 2018*; *Summers et al., 2015*; *Wilson et al., 2017*).

The involvement of ASE neurons, largely described as gustatory neurons, in the avoidance of *E. faecalis* (*Figure 3A,B*) suggests that taste may play a role in pathogen avoidance in *C. elegans* in certain contexts. ASE neurons have also been shown to be carbon dioxide sensors (*Bretscher et al., 2011*), and polymodal sensory neurons are common in *C. elegans* (*Kaplan and Horvitz, 1993*; *Metaxakis et al., 2018*), raising the possibility that ASE sensation of carbon dioxide, and not gustation, may elicit avoidance of *E. faecalis*. Additionally, a recent study implicated ASEL as a secondary sensory neuron in the detection of the food odor benzaldehyde (*Leinwand et al., 2015*). The results of the olfactory aversive learning experiments, however, suggest that ASE neurons are not involved in the olfactory component of *E. faecalis* avoidance. Whether they are involved in some other learning process underpinning avoidance remains unclear. Interestingly, AWC neurons are also recruited into a salt-sensing circuit via neuropeptide signaling from ASE neurons depending on salt concentration changes (*Leinwand and Chalasani, 2013*), leaving open the possibility that AWC neurons also play a role in avoidance of *E. faecalis* (*Figure 3A,B*) not only through olfactory aversive learning (*Figure 6C–E*) but also through taste aversive learning. Further studies will need to be conducted to determine which possible sensory modality is important for ASE-dependent and AWC-dependent avoidance of *E. faecalis*. ASE neurons were also identified as part of the complex circuitry underlying the decision to leave a resource-depleted food patch, possibly via carbon dioxide sensing (*Milward et al., 2011*), suggesting an interesting link between pathogen avoidance and adaptive food leaving.

AWB and AWC involvement in odor preference and olfactory aversive learning is well established in various contexts (*Ha et al., 2010*; *Harris et al., 2014*; *Yoshida et al., 2012*), and the present study extends this to *C. elegans* interactions with *E. faecalis* (*Figure 6*) while providing direct evidence of their involvement in avoidance for both *E. faecalis* and *P. aeruginosa* (*Figure 5B*). The olfactory circuitry in *C. elegans* has been described in detail (*Chalasani et al., 2007*; *Leinwand et al., 2015*). AWC neurons also communicate and receive feedback from other interneurons through the use of neuropeptides, such as the NLP-1-NPR-11-INS-1 feedback loop between AWC and AIA interneurons (*Chalasani et al., 2010*). The circuitry for olfactory aversive learning has also received extensive study in the context of *P. aeruginosa* with serotonin and various neuropeptides playing crucial roles in learning and foraging states (*Chen et al., 2013*; *Fadda et al., 2020*; *Flavell et al., 2013*; *Ha et al., 2010*; *Harris et al., 2014*; *Zhang et al., 2005*). Interestingly, both AWB and AWC neurons were previously shown to play a role in avoidance of normally attractive food following undernourishment (*Olofsson, 2014*), providing another link between pathogen avoidance behavior and foraging. Furthermore, in a recent study, the transcription factor Nrf2/*skn-1* was shown to be required in the AIY interneuron for the integration of information from ASE and AWC neurons during foraging (*Wilson et al., 2017*). Whether this same AIY neuronal integration is required for pathogen avoidance remains to be seen. Previous work in *Drosophila* has also revealed olfactory-mediated avoidance of harmful microbes (*Stensmyr et al., 2012*), illustrating that similar mechanisms are at play across different species.

The discovery of *gon-2* and *gtl-2* playing a role in pathogen avoidance is novel. The most well-known impact of loss of *gon-2* function is severe impairment of gonadogenesis due to disruption of gonadal cell divisions (*Sun and Lambie, 1997*). This impairment is most likely responsible for the results observed for avoidance of *P. aeruginosa* (*Figure 7A,B*), as the germline and associated tissues are necessary for this avoidance behavior (*Kaletsky et al., 2020*; *Moore et al., 2019*; *TeKippe and Aballay, 2010*). The results of the *glp-1* experiments (*Figure 7A,B*) further support this idea. However, the germline seems to play no role in avoidance of *E. faecalis*, and therefore other functions of *gon-2* and *gtl-2* may be responsible. The germline plays a role in a transgenerational learned avoidance of *P. aeruginosa* (*Moore et al., 2019*), but the lack of involvement of the germline (*Figure 7A,B*), bacterial sRNAs (*Figure 1E*), and ASI neurons (*Figure 5B*) in avoidance of *E. faecalis* suggests that such a process may not exist in the case of *E. faecalis* infection. Because *gon-2*

is known to be highly expressed in the intestine, where it is responsible for electrolyte homeostasis (*Teramoto et al., 2005*), and rescue of *gon-2* intestinal expression using the *ges-1* promoter rescues *E. faecalis* avoidance phenotypes, it may function as an intestinal receptor for the changes elicited by *E. faecalis* colonization. Magnesium excretion, and perhaps additional electrolytes, requires the activity of *gtl-2* in the excretory cell (*Teramoto et al., 2010*), and rescue of *gtl-2* excretory cell expression using the *sulp-4* promoter rescues *E. faecalis* avoidance phenotypes. Thus, it is possible that the role of *gon-2* and *gtl-2* in pathogen avoidance is related to sensation of an electrolyte perturbance caused by intestinal distention. Both *gon-2* and *gtl-1*, though not *gtl-2*, are also required for maintaining the rhythm of the *C. elegans* defecation motor program (*Kwan et al., 2008*), which could also result in aberrant avoidance behaviors, though loss of *gtl-1* does not seem to affect avoidance (*Figure 7—figure supplement 1*). Finally, these TRPM channels could also be mechano-nociceptors, as described in other animal models (*Mueller-Tribbensee et al., 2015*), that sense intestinal distention directly, though the data presented here could also be consistent with a less direct role as described above. Indeed, TRPM channels display many distinct functions in various settings, including the absorption of calcium and magnesium ions by TRPM6 and TRPM7 in the human colon, TRPM7-mediated pacemaker activity of interstitial cells of Cajal, and colonic nociception by mouse TRPM8 (*Holzer, 2011*; *Mueller-Tribbensee et al., 2015*). Thus, it remains unclear how exactly GON-2 and GTL-2 function in pathogen avoidance. Because the intestine is not directly innervated, any signal coming from it must be extra-synaptic, such as an intestinal neuropeptide (*Lee and Mylonakis, 2017*). All animals are under pressure to develop behaviors that allow them to flee potential pathogens (*Sarabian et al., 2018*). Future work will continue to elucidate the mechanisms underlying avoidance of pathogenic threats.

## Materials and methods

### Key resources table

| Reagent type (species) or resource | Designation | Source or reference | Identifiers | Additional information |
|---|---|---|---|---|
| Strain, strain background (*Escherichia coli*) | OP50 | *Caenorhabditis* Genetics Center (CGC) | OP50 | |
| Strain, strain background (*E. coli*) | OP50-GFP | CGC | OP50-GFP | |
| Strain, strain background (*E. coli*) | HT115 | Source BioScience | HT115 | |
| Strain, strain background (*Pseudomonas aeruginosa*) | PA14 | Frederick M. Ausubel laboratory | PA14 | |
| Strain, strain background (*P. aeruginosa*) | PA14-GFP | Frederick M. Ausubel laboratory | PA14-GFP | |
| Strain, strain background (*Enterococcus faecalis*) | OG1RF | Danielle A. Garsin laboratory | OG1RF | |
| Strain, strain background (*E. faecalis*) | OG1RF-GFP | Danielle A. Garsin laboratory | OG1RF-GFP | |
| Strain, strain background (*E. faecalis*) | OG1RF ΔfsrB | Danielle A. Garsin laboratory | OG1RF ΔfsrB | |
| Strain, strain background (*E. faecium*) | E007 | Danielle A. Garsin laboratory | E007 | |

*Continued on next page*

*Continued*

| Reagent type (species) or resource | Designation | Source or reference | Identifiers | Additional information |
|---|---|---|---|---|
| Strain, strain background (*Staphylococcus aureus*) | NCTC8325 | National Collection of Type Cultures | NCTC8325 | |
| Strain, strain background (*Caenorhabditis elegans*) | N2 Bristol | CGC | N2 | |
| Strain, strain background (*C. elegans*) | npr-1(ad609) | CGC | DA609 | |
| Strain, strain background (*C. elegans*) | npr-1(ok1447) | CGC | RB1330 | |
| Strain, strain background (*C. elegans*) | ocr-2(ak47) | CGC | CX4544 | |
| Strain, strain background (*C. elegans*) | osm-9(ky10) | CGC | CX10 | |
| Strain, strain background (*C. elegans*) | gcy-35(ok769); npr-1(ad609) | Aballay laboratory | | |
| Strain, strain background (*C. elegans*) | tax-2(p694); npr-1(ad609) | Aballay laboratory | | |
| Strain, strain background (*C. elegans*) | tax-4(p678); npr-1(ad609) | Aballay laboratory | | |
| Strain, strain background (*C. elegans*) | tax-2(p671) | CGC | PR671 | |
| Strain, strain background (*C. elegans*) | tax-2(p694) | CGC | PR694 | |
| Strain, strain background (*C. elegans*) | tax-4(p678) | CGC | PR678 | |
| Strain, strain background (*C. elegans*) | tax-2(p694); lin-15 and lin-15A(n765); dbEx723[gcy-32p::tax-2(cDNA)::SL2::GFP + lin-15(+)] | CGC | AX2159 | |
| Strain, strain background (*C. elegans*) | tax-2(p694); lin-15 and lin-15A(n765); dbEx724[flp-6p::tax-2(cDNA)::SL2::GFP + lin-15(+)] | CGC | AX2161 | |
| Strain, strain background (*C. elegans*) | che-1(p680) | CGC | PR680 | ASE(−) |
| Strain, strain background (*C. elegans*) | peIs1713 [sra-6p::mCasp-1+unc-122p::mCherry] | CGC | JN1713 | ASH(−) |
| Strain, strain background (*C. elegans*) | oyIs84 [gpa-4p::TU#813 + gcy-27p::TU#814 + gcy-27p::GFP + unc-122p::DsRed] | CGC | PY7505 | ASI(−) |

*Continued on next page*

*Continued*

| Reagent type (species) or resource | Designation | Source or reference | Identifiers | Additional information |
|---|---|---|---|---|
| Strain, strain background (*C. elegans*) | peIs1715 [str-1p::mCasp-1+unc-122p::GFP] | CGC | JN1715 | AWB(−) |
| Strain, strain background (*C. elegans*) | oyIs85 [ceh-36p::TU#813 + ceh-36p::TU#814 + srtx-1p::GFP + unc-122p::DsRed] | CGC | PY7502 | AWC(−) |
| Strain, strain background (*C. elegans*) | agIs26 [clec-60p::GFP + myo-2p::mCherry] | CGC | JIN810 | |
| Strain, strain background (*C. elegans*) | unc-119(ed3); eEx650 [ilys-3p::GFP + unc-119(+)] | CGC | CB6710 | |
| Strain, strain background (*C. elegans*) | pmk-1(km25) | CGC | KU25 | |
| Strain, strain background (*C. elegans*) | fshr-1(ok778) | CGC | RB911 | |
| Strain, strain background (*C. elegans*) | bar-1(ga80) | CGC | EW15 | |
| Strain, strain background (*C. elegans*) | nol-6(ac1) | CGC | AY1 | |
| Strain, strain background (*C. elegans*) | gtl-2(n2618) | CGC | CZ9957 | |
| Strain, strain background (*C. elegans*) | gon-2(q362) | CGC | EJ26 | |
| Strain, strain background (*C. elegans*) | gon-2(q388) | CGC | EJ1158 | |
| Strain, strain background (*C. elegans*) | glp-1(e2141) | CGC | CB4037 | |
| Strain, strain background (*C. elegans*) | gtl-1(ok375) | CGC | VC244 | |
| Strain, strain background (*C. elegans*) | gtl-2(tm463) | CGC | LH202 | |
| Strain, strain background (*C. elegans*) | trpa-1(ok999) | CGC | RB1052 | |
| Strain, strain background (*C. elegans*) | trpa-2(ok3189) | CGC | TQ233 | |
| Strain, strain background (*C. elegans*) | gon-2(q388); gon-2p::gon-2(cDNA)::SL2::GFP | This Study | AY157 | *gon-2* expression, own promoter |
| Strain, strain background (*C. elegans*) | gon-2(q388); ges-1p::gon-2(cDNA)::SL2::GFP | This Study | AY158 | *gon-2* expression, intestine specific promoter |

*Continued on next page*

*Continued*

| Reagent type (species) or resource | Designation | Source or reference | Identifiers | Additional information |
|---|---|---|---|---|
| Strain, strain background (*C. elegans*) | *gtl-2(n2618); gtl-2p::gtl-2(cDNA)::SL2::GFP* | This Study | AY159 | *gtl-2* expression, own promoter |
| Strain, strain background (*C. elegans*) | *gtl-2(n2618); sulp-4p::gtl-2(cDNA)::SL2:GFP* | This Study | AY160 | *gtl-2* expression excretory cell promoter |
| Software, algorithm | GraphPad Prism 8 | GraphPad Software | | https://www.graphpad.com/scientific-software/prism/ |
| Software, algorithm | ImageJ | NIH | | https://imagej.nih.gov/ij/ |
| Software, algorithm | Leica LAS v4.6 | Leica | | https://www.leica-microsystems.com/ |
| Other | Hypoxia chamber | STEMCELL Technologies | CAT# 27310 | |

## Bacterial strains

The following bacterial strains were used: *Enterococcus faecalis* OG1RF, *E. faecalis* OG1RF Δ*fsrB*, *E. faecalis* OG1RF-GFP, *E. faecium* E007, *Escherichia coli* OP50, *E. coli* OP50-GFP, *E. coli* HT115(DE3), *Pseudomonas aeruginosa* PA14, *P. aeruginosa* PA14-GFP, and *Staphylococcus aureus* NCTC8325. *E. coli* and *P. aeruginosa* bacterial strains were grown in Luria-Bertani (LB) broth at 37°C, while the rest were grown in brain-heart infusion (BHI) broth at 37°C. All sources are listed in the Key Resources Table.

## *C. elegans* strains and growth conditions

*C. elegans* hermaphrodites were maintained on *E. coli* at 20°C unless otherwise indicated. Bristol N2 was used as the wild-type control unless otherwise indicated. All other strains and their sources are listed in the Key Resources Table.

## Construction of transgenic strains

The following transgenic lines were generated in this study:

gon-2(q388);gon-2p::gon-2(cDNA)::SL2::GFP
gon-2(q388);ges-1p::gon-2(cDNA)::SL2::GFP
gtl-2(tm1463);gtl-2p::gtl-2(cDNA)::SL2::GFP
gtl-2(tm1463);sulp-4p::gtl-2(cDNA)::SL2::GFP

The transgenic strains were generated by micro-injecting the DNA of an expression plasmid (50 ng/μL for the first three and 5 ng/μL for the last), along with a co-injection marker plasmid (*coel::RFP*, 50 ng/μL). The plasmids were maintained as extrachromosomal arrays. The expression plasmids consisted of a promoter region and full-length cDNA subcloned into pPD95.77 (Fire Lab *C. elegans* Vector Kit; Addgene) containing an SL2::GFP region (*Cao et al., 2017*) via *SphI-XmaI* restriction sites. The *gon-2* promoter region used was 3.5 kbp upstream from the *gon-2* gene, the *ges-1* promoter region used was 3.3 kbp upstream from the *ges-1* gene, the *gtl-2* promoter region used was 2.9 kbp upstream from the start codon of F54D1.5, and the *sulp-4* promoter was 4.4 kbp upstream from the start codon of K12G11.1. Full-length *gon-2* and *gtl-2* cDNA were 6099 and 4221 bp, respectively.

## RNA interference

RNAi was used to generate loss-of-function RNAi phenotypes by feeding nematodes *E. coli* strain HT115(DE3) expressing double-stranded RNA (dsRNA) homologous to a target gene (*Fraser et al., 2000*; *Timmons and Fire, 1998*). RNAi was carried out as described previously (*Singh and Aballay, 2017*). Briefly, *E. coli* with the appropriate vectors were grown in LB broth containing ampicillin (100 μg/mL) and tetracycline (12.5 μg/mL) at 37°C overnight and plated onto NGM plates containing 100

µg/mL ampicillin and 3 mM isopropyl β-ᴅ-thiogalactoside (RNAi plates). RNAi-expressing bacteria were allowed to grow for 2 days at 20℃. Gravid adults were transferred to RNAi-expressing bacterial lawns and allowed to lay eggs for 5 hr. The gravid adults were removed, and the eggs were allowed to develop at 20℃ to young adults for subsequent assays. The RNAi clones were from the Ahringer RNAi library.

## Lawn avoidance assays

Bacterial cultures were grown by inoculating individual bacterial colonies into 2 mL of either LB or BHI broth and growing them for 5–6 hr on a shaker at 37℃. Then, 20 µL of the culture was plated onto the center of 3.5-cm-diameter BHI or standard slow-killing (SK) plates (modified NGM agar plates [0.35% instead of 0.25% peptone]) as indicated. The plates were then incubated overnight at 37℃. The plates were cooled to room temperature for at least 30 min before seeding with animals. Synchronized young gravid adult hermaphroditic animals grown on *E. coli* OP50 were transferred outside the indicated bacterial lawns, and the numbers of animals on and off the lawns were counted at the specified times for each experiment. Three 3.5-cm-diameter plates were used per trial in every experiment. Occupancy index was calculated as ($N_{on}$ lawn/$N_{total}$). For first exiting events, individual animals were monitored and the time that the animal first left the lawn was recorded. For exploratory events, entry and exiting events were recorded for 10 min at 50 min and 3 hr and 50 min after transfer.

## Avoidance assays at 8% oxygen

Avoidance assays as described above were carried out in a hypoxia chamber. Briefly, after young gravid adult hermaphroditic animals were transferred to the avoidance plates, the plates were placed in the hypoxia chamber and the lids of the plates were removed. The chamber was purged with 8% oxygen (balanced with nitrogen) for 5 min at a flow rate of 25 L/min. The chamber was then sealed, and assays were carried out. Control plates were incubated at ambient oxygen.

## Avoidance assays with bacterial RNA

RNA from bacterial pellets was isolated and used for avoidance assays as previously described (*Kaletsky et al., 2020*). Briefly, bacteria for RNA collection were grown on SK or BHI plates overnight at 37℃. Bacterial lawns were collected from the surface of the plates using 1 mL of M9 buffer and a cell scraper. The resulting suspension was transferred to a 15 mL conical tube. PA14, OG1RF, or OP50 from 15 lawns was pooled in each tube and pelleted at 5000 g for 10 min at 4℃. The supernatant was discarded, and the pellet was resuspended in 1 mL of Trizol LS for every 100 µL of bacterial pellet recovered. The pellet was resuspended by vortexing and subsequently frozen at −80℃ until RNA isolation. Two hundred and forty micrograms of total RNA was placed directly onto OP50 lawns and allowed to dry at room temperature before transferring worms over for avoidance assays.

## Aversive training

Training plates of 3.5 cm diameter containing either *E. coli* OP50 on SK agar or *E. faecalis* OG1RF on BHI agar were made as described above for avoidance assays. For RNA training assays, SK plates containing *E. coli* OP50 were spotted with the appropriate isolated RNA. Young gravid adult hermaphroditic animals grown on *E. coli* OP50 were transferred to the training plates and allowed to roam for 4 hr for *E. faecalis* training or 24 hr for RNA training. They were then transferred to the appropriate assay plates.

## Two-choice preference assays

Bacterial cultures were grown as indicated in the lawn avoidance assays above. Then, 20 µL of each inoculum was plated on opposite sides of a 6-cm-diameter BHI or SK plate and incubated overnight at 37℃. The plates were cooled to room temperature for at least 30 min before seeding with animals. For the 'Paralyzed' condition, 1 µL of 1M sodium azide was spotted onto each bacterial lawn. Young gravid adult hermaphroditic animals grown on *E. coli* OP50 were transferred to the center of plates equidistant from both the lawns. The numbers of animals on both lawns were counted at the specified times for each experiment. Three 6-cm-diameter plates were used per trial in every experiment. The *E. faecalis* choice index (*E. faecalis* CI) was calculated as follows:

$$E.faecalis\ \text{CI} \frac{[(\text{No. of worms on } E.faecalis) - (\text{No. of worms on } E.coli)]}{[(\text{No. of worms on } E.faecalis) + (\text{No. of worms on } E.coli)]}$$

Choice indices to other bacteria were similarly calculated.

### Two-choice odor preference assays (lid choice assays)

A modified version of the two-choice preference assays was carried out as previously described (*Worthy et al., 2018*) to assess odor preference alone. To do this, bacterial cultures were grown and plated as indicated in the two-choice preference assays above. Then, agar plugs with the bacterial lawns were cut from the plates and transferred to the lids of new 6-cm-diameter BHI agar plates without any bacteria, so that the bacteria on the agar plugs faced down toward the plate surface on opposite sides of the plate. Two microliters of 1 M sodium azide was spotted on the surface below each agar plug. Young gravid adult hermaphroditic animals grown on *E. coli* OP50 (naïve) or from *E. faecalis* OG1RF training plates were transferred to the center of plates equidistant from both agar plugs. The numbers of animals paralyzed under each agar plug were counted after 1 hr. Three 6-cm-diameter plates were used per trial in every experiment. The choice index was calculated as indicated above.

### Dry-drop assay

The dry-drop assay was carried out as previously described (*Tran et al., 2017*). Using a capillary, a dry drop of either SDS (positive control), BHI or LB (negative control), or *E. faecalis* or *E. faecium* cultures was placed on a dry SK plate in front of a forward-moving animal. A response was counted if an animal initiated backward movement. Response Index = number of responses/total number of drops.

### Imaging and quantification

Fluorescence imaging was carried out as described previously (*Singh and Aballay, 2017*) with slight modifications. Briefly, the animals were anesthetized using an M9 salt solution containing 50 mM sodium azide and mounted onto 2% agar pads. The animals were then visualized using a Leica M165 FC fluorescence stereomicroscope. For quantification of intestinal lumen distention, brightfield images were acquired at each time point using the Leica LAS v4.6 software, and the diameter of the intestinal lumen was measured using ImageJ software. For quantification of fluorescent immune reporters, fluorescent images were acquired using the Leica LAS v4.6 software in grayscale as presented, and the fluorescence intensity was measured and averaged across three points in the intestine of each animal at the indicated time points using ImageJ software.

### Statistical analysis

The statistical analysis was performed with Prism 8 (GraphPad). For box-and-whisker plots, the center line depicts the median, and the box range depicts the first and third quartile, while the whiskers depict the minimum and maximum data points. For all box plots, individual dots represent individual trials. For time course figures, the mean and standard deviation of nine trials is depicted at each time point. All experiments were performed in triplicate on three separate days. Unpaired t-tests, one-way or two-way ANOVA with subsequent group comparisons were performed as indicated in the figure legends. In the figures, ns denotes not significant, and asterisks (*) denote statistical significance as follows: *$p \leq 0.05$; **$p \leq 0.01$; ***$p \leq 0.001$; ****$p \leq 0.0001$, as compared with the appropriate controls.

## Acknowledgements

We thank Danielle A Garsin (McGovern Medical School) for providing the *E. faecalis* OG1RF *ΔfsrB* and GFP strains. Some strains used in this study were provided by the Caenorhabditis Genetics Center (CGC), which is funded by the NIH Office of Research Infrastructure Programs (P40OD01044). Images were created using Microsoft PowerPoint and BioRender.com.

## Additional information

### Funding

| Funder | Grant reference number | Author |
| --- | --- | --- |
| National Institutes of Health | GM070977 | Alejandro Aballay |
| National Institutes of Health | AI156900 | Alejandro Aballay |

The funders had no role in study design, data collection and interpretation, or the decision to submit the work for publication.

### Author contributions

Adam Filipowicz, Conceptualization, Data curation, Formal analysis, Validation, Investigation, Visualization, Methodology, Writing - original draft, Writing - review and editing; Jonathan Lalsiamthara, Investigation, Methodology, Writing - review and editing; Alejandro Aballay, Conceptualization, Data curation, Formal analysis, Supervision, Funding acquisition, Investigation, Writing - original draft, Project administration, Writing - review and editing

### Author ORCIDs

Adam Filipowicz (iD) https://orcid.org/0000-0003-4458-1073
Alejandro Aballay (iD) https://orcid.org/0000-0002-5975-3352

### Decision letter and Author response

Decision letter https://doi.org/10.7554/eLife.65935.sa1
Author response https://doi.org/10.7554/eLife.65935.sa2

## Additional files

### Supplementary files

• Transparent reporting form

### Data availability

All data generated or analyzed during this study are included in the manuscript and supporting files.

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
