## [Decision Letter]

**Acceptance summary:**

Understanding how animals learn to avoid pathogens is an important question that bridges physiology, neuroscience, and immunity. Using the nematode *C. elegans*, your work identifies a fascinating mechanism that links ingestion of the pathogen *E. faecalis* to intestinal distention (possibly detected by TRPM channels) and subsequent aversive learning. Your results show that *C. elegans* uses different strategies to avoid different pathogens and will be of broad interest to researchers interested in pathogen resistance mechanisms, sensory behavior, and the intersection of these processes.

**Decision letter after peer review:**

Thank you for submitting your article "TRPM channels mediate learned pathogen avoidance following intestinal distention" for consideration by *eLife*. Your article has been reviewed by 3 peer reviewers, including Douglas Portman as the Reviewing Editor and Reviewer #1, and the evaluation has been overseen by Piali Sengupta as the Senior Editor. The following individual involved in review of your submission has agreed to reveal their identity: Michael O'Donnell (Reviewer #2).

Essential revisions:

After discussing their evaluations, the reviewers feel that several points must be addressed in a revised manuscript.

1. All three reviewers felt that the evidence that intestinal bloating drives pathogen avoidance learning was premature and largely correlative.

a. At a minimum, more needs to be known about the site of action of gon-2 and/or gtl-2.

b. Additionally, the manuscript would be significantly improved by demonstrating that anterior bloating has an instructive role in learning, perhaps by finding way (beyond the use of nol-6 and aex-5) to trigger bloating in the absence of pathogen ingestion. Otherwise, the text will need to be revised to make it clear that the proposed role of bloating is speculative.

2. The reviewers are concerned that the role of npr-1 in *E. faecalis* avoidance is overstated and potentially misleading. As explained in more detail below, the available evidence suggests that the models in 5E and 7E are not the simplest explanation of the data. Specifically, the possibility that npr-1 is not a part of the *E. faecalis* avoidance pathway per se, but rather is a separate input that can modulate this behavior when animals are in the presence of high ambient O2, should be acknowledged in the text and figures.

3. While the paper implies that the behavioral plasticity you observe is associative learning, this has not been clearly demonstrated. The paper would be significantly improved by carrying out additional behavioral studies to determine whether associative learning is indeed taking place, as described by reviewer #2 below. Alternatively, this issue can be handled by modifying the text to be clear about the nature of the behavioral change triggered by *E. faecalis* exposure.

*Reviewer #1 (Recommendations for the authors):*

1. The claim of the title, that "TRPM channels mediate learned pathogen avoidance following intestinal distention," is not well supported. The evidence that intestinal distention is the trigger for learning is almost completely correlative. A convincing demonstration of causality, beyond the evidence provided by the nol-6 and aex-5 experiments, would significantly improve the paper.

2. The role of the two TRPM channels identified is only superficially probed. While it seems clear that their function is required for E.f. avoidance, it is not clear that their function is necessary for learning, much less that they "mediate" learning in response to intestinal distention. This could be improved by determining the site of action of these factors, determining whether they have a specific role in pathogen learning and olfactory plasticity (as opposed to baseline responses to a bacterial lawn), and asking if they are specifically activated by intestinal distention (though I'm not quite sure how this might be done).

3. The role of npr-1 in pathogen avoidance is overstated, if not misconstrued. As it stands, the paper shows simply that, for npr-1(lf) animals, high O2 is a more potent aversive stimulus than is E.f. This is fine, but I don't see how it leads to the idea that npr-1 activity is downstream of pathogen exposure or detection, as is indicated in Figures5E and 7E, or that pathogen avoidance is "regulated" by npr-1, as stated in the first sentence of the Discussion.

*Reviewer #2 (Recommendations for the authors):*

As I described in the public comments, this is a really excited set of findings and if the hypotheses are fully supported, I think this would result in a publication of very broad interest.

The main concerns I have are related to the strength and specificity of the assertions made in each of the major finding areas I described. They are repeated below along with some suggestions that I think could strengthen these arguments.

1. Anterior intestinal distension causes rapid behavioral avoidance.

I found the authors' previous demonstrations linking distension to behavioral avoidance in *Pseudomonas* quite impressive and in my opinion the experiments using nol-6 to disrupt distension were particularly important in establishing that. In this case, because the behavioral avoidance is so rapid, I think demonstrating causality is even more challenging and important, if you intend to assert this as strongly as is currently written. Given its role in peptide processing, I'd suggest that solely using aex-5 mutations to establish sufficiency is not ideal here.

I recognize that there aren't currently any tools available to block bloating via *E. faecalis* since nol-6 RNAi does not, but more carefully comparing the temporal progression of bloating relative to behavioral changes might be the next best approach. Lawn avoidance behavior is so rapid – is it really plausible that bloating might occur in minutes? Perhaps you could measure the intestine of animals that rapidly avoid compared to those that do not to see if there is a bias? Colonization was visualized at 4hr but is suggested to cause distension, which of these occurs first?

Also, it might be that certain components of behavioral avoidance are driven by physical bloating and others are not. For example, ASE, AWC and AWB are all involved in lawn avoidance but ASE is not needed for olfactory choice.

Additionally, to more strongly rule out immune signaling, would it not be more definitive to use fshr-1, pmk-1 or bar-1 mutants?

2. Avoidance occurs via an aversive learning process.

The olfactory plasticity data shown are quite strong and convincing. My main concern is the specificity of the description of learning in this case. In my opinion, aversive learning is typically used as a shorthand to describe an associative learning paradigm. I assume the authors are not proposing some form of non-associative learning such as habituation or sensitization. For the olfactory learning process to be associative, there should be specificity to the response to odors that were present during training. In my opinion, a change in olfactory preference from *E. faecalis* to *E. coli* doesn't directly demonstrate this specificity.

One way to address this would be to pair an AWB or AWC-sensed odor with the bacteria during training, then test if responses to this odor is specifically altered. If it turns out that this is not associative learning, this would still be quite interesting as it might indicate there are mechanisms to generally downregulate attraction to distension-causing bacteria – but this type of scenario should probably be referred to as some sort of state-dependent plasticity, as opposed to learning. If the effect is associative, can you rule out a satiety effect? Are the AWC/AWC dependent behaviors related to learning while the ASE effects are not? This concern might be solved by altering the text or interpretations.

3. Intestinal gon-2 and gtl-2 may sense intestinal bloating.

In my opinion, this section represents potentially the most exciting finding but also the area in which the data are currently least strong. While you haven't directly stated that these channels are intestinal bloating sensors, this is strongly implied and/or suggested throughout the manuscript and impact statements.

While gon-2 and gtl-2 are strongly expressed in the intestine, they are also expressed in other tissues, which might include neurons. To establish their site of action they should perform tissue specific rescue experiments.

Are the intestines of gon-2/gtl-2 mutants distended when grown on *E. faecalis* or is colonization affected?

If these channels act in the intestine, short of directly demonstrating a sensory function, it seems reasonable that they could play many different roles, which should probably be acknowledged in the text.

*Reviewer #3 (Recommendations for the authors):*

The title of this paper emphasizes their discovery of TRPM channels in this learned pathogen avoidance response. I thought their characterization of the neural circuitry involved is equally important and could be emphasized instead in the title. To further characterize the TRPM channels, the authors could consider demonstrating that GON2 and GTL-2 function in the intestine (through intestinal expression in the mutant background), which would support their hypothesis nicely. Alternatively, the authors could further implicate these loci by rescuing the avoidance phenotype by expressing these genes under their own promoters in the mutant background.

---

## [Author Response]

Essential revisions:After discussing their evaluations, the reviewers feel that several points must be addressed in a revised manuscript.1. All three reviewers felt that the evidence that intestinal bloating drives pathogen avoidance learning was premature and largely correlative.a. At a minimum, more needs to be known about the site of action of gon-2 and/or gtl-2.

We agree with these criticisms and have thus carried out additional experiments to address these issues. To learn more about the site of action of *gon-2* and *gtl-2* we generated the following rescue strains: *gon-2(q388); gon-2p::gon-2(cDNA)::SL2::GFP, gon-2(q388); ges1p::gon-2(cDNA)::SL2::GFP, gtl-2(n2618); gtl-2p::gtl-2(cDNA)::SL2::GFP,* and *gtl-2(n2618); sulp-4p::gtl-2(cDNA)::SL2:GFP*. The *ges-1* promoter drives expression in the intestine while the *sulp-4* promoter drives expression in the excretory cell. We used these strains in *E. faecalis* avoidance assays and found that the expression of *gon-2* and *gtl-2* rescued the avoidance defects (New Figure 7C and D). The manuscript now reads:

“In addition to gonadal expression, gon-2 is known to exhibit intestinal expression (Teramoto et al., 2005). […] Together, these results indicate that gon-2 expression in the intestine and gtl-2 expression in the excretory cell is sufficient for *C. elegans* avoidance of *E. faecalis*.”

We also confirmed that *gon-2* and *gtl-2* mutants displayed intestinal distention following exposure to *E. faecalis* (New Figure 8 – Supplement 1) and that these animals have impaired olfactory aversive learning (New Figure 8C and D). The manuscript now reads:

“Because gon-2(lf) and gtl-2(lf) animals display wild-type levels of anterior intestinal distention (Figure 8—figure supplement 1), we hypothesized that the effect observed in these animals was due to an inability to sense intestinal distention either directly or indirectly […] Altogether, these findings indicate that GON-2 and GTL-2 are required for the learned avoidance of *E. faecalis*, though their exact mechanistic role remains to be determined (Figure 8E).”

b. Additionally, the manuscript would be significantly improved by demonstrating that anterior bloating has an instructive role in learning, perhaps by finding way (beyond the use of nol-6 and aex-5) to trigger bloating in the absence of pathogen ingestion. Otherwise, the text will need to be revised to make it clear that the proposed role of bloating is speculative.

As appreciated by the reviewers, it is not possible to trigger anterior intestinal distention in the absence of pathogen ingestion, and have modified the manuscript to make this clear. The title for this section now reads: “Avoidance of *E. faecalis* follows anterior intestinal distention and is independent of virulence”.

However, following the suggestion of Reviewer 2, we took a more detailed look at the temporal relationship between distention and avoidance. To do this, animals were placed on avoidance assay plates with *E. faecalis* and then monitored for when they first left the lawn. These animals were then immediately transferred to agarose pads for imaging, along with animals that were still on the lawn. The anterior intestinal diameter of both groups of animals was then measured. Importantly, animals outside the lawn had much larger anterior intestines (New Figure 2 – Supplement 1). This supports the notion that anterior intestinal distention triggers flight from the lawn. The manuscript now reads:

“Because we first measured the anterior intestinal diameter at 1 hour, it was not clear whether distention preceded avoidance or vice versa. […] It is therefore likely that fast accumulation of *E. faecalis* in the anterior intestine induces an aversive response to the pathogen, leading to avoidance.”

2. The reviewers are concerned that the role of npr-1 in *E. faecalis* avoidance is overstated and potentially misleading. As explained in more detail below, the available evidence suggests that the models in 5E and 7E are not the simplest explanation of the data. Specifically, the possibility that npr-1 is not a part of the *E. faecalis* avoidance pathway per se, but rather is a separate input that can modulate this behavior when animals are in the presence of high ambient O2, should be acknowledged in the text and figures.

We agree these criticisms and find the reviewer’s explanation of the data much more compelling. The reference to NPR-1 in the title of this section has been removed, and the language has been changed throughout the manuscript to reflect that NPR-1 regulated hyperoxia avoidance opposes avoidance of *E.f* but does not necessarily fit into the avoidance pathway. This section now reads:

“Furthermore, when animals were taken out of the low oxygen environment and allowed to roam at atmospheric oxygen levels for one-hour, wild-type animals remained off the *E. faecalis* lawns while both npr-1(lf) strains migrated back onto the lawns (Figure 4C) indicating that hyperoxia avoidance is a more potent aversive stimulus than *E. faecalis* in npr-1(lf) animals.”

In addition, the models have been modified as suggested (New Figures 5E and 8E).

3. While the paper implies that the behavioral plasticity you observe is associative learning, this has not been clearly demonstrated. The paper would be significantly improved by carrying out additional behavioral studies to determine whether associative learning is indeed taking place, as described by reviewer #2 below. Alternatively, this issue can be handled by modifying the text to be clear about the nature of the behavioral change triggered by *E. faecalis* exposure.

We thank Reviewer 2 for this critique and the suggested experiments, which we have performed. Specifically, we paired the attractive odorant benzaldehyde with *E. faecalis* and subsequently carried out odor chemotaxis assays with these trained animals, as well as naïve animals, to test if a negative association had formed during training (New Figure 8C). The data shows that wild-type animals do form a negative association, while both naïve and trained animals lacking AWC neurons show no attraction. Interestingly, naïve animals lacking AWB had the same attraction to benzaldehyde as wild-type animals, but trained animals lacking AWB showed no associative learning (New Figure 8D and E). The manuscript now reads:

“To strengthen the idea that avoidance of *E. faecalis* involves aversive learning, we paired an attractive odorant, benzaldehyde, with lawns of *E. faecalis* and exposed animals to this pairing for four hours (Figure 6C). […] Interestingly, animals with ablated AWB neurons displayed wild-type levels of naïve attraction but did not show a decrease in attraction upon training, indicating that AWB neurons may be required for this aversive learning process.”

Reviewer #1 (Recommendations for the authors):1. The claim of the title, that "TRPM channels mediate learned pathogen avoidance following intestinal distention," is not well supported. The evidence that intestinal distention is the trigger for learning is almost completely correlative. A convincing demonstration of causality, beyond the evidence provided by the nol-6 and aex-5 experiments, would significantly improve the paper.

As explained in our responses to Essential Revisions, we modified the manuscript and performed additional experiments to address this criticism. The revised manuscript now has the following title for this section: “Avoidance of *E. faecalis* follows anterior intestinal distention independent of virulence.”

We were able to more convincingly show the temporal relationship between distention and avoidance (New Figure 2 – Supplement 1). This finding supports the notion that distention triggers flight away from the bacterial lawn, as most of the animals which had just left the lawn showed distended anterior intestines, while those still inside the lawn did not. The manuscript now reads:

“Because we first measured the anterior intestinal diameter at 1 hour, it was not clear whether distention preceded avoidance or vice versa. […] It is therefore likely that fast accumulation of *E. faecalis* in the anterior intestine induces an aversive response to the pathogen, leading to avoidance.”

2. The role of the two TRPM channels identified is only superficially probed. While it seems clear that their function is required for E.f. avoidance, it is not clear that their function is necessary for learning, much less that they "mediate" learning in response to intestinal distention. This could be improved by determining the site of action of these factors, determining whether they have a specific role in pathogen learning and olfactory plasticity (as opposed to baseline responses to a bacterial lawn), and asking if they are specifically activated by intestinal distention (though I'm not quite sure how this might be done).

We have determined the site of action by using rescue constructs as described in Essential Revisions and carrying out avoidance assays with these strains (New Figures 7C and D). This revealed that expression of *gon-2* in intestine and *gtl-2* in the excretory cell rescued the avoidance phenotype. The revised manuscript reads:

“In addition to gonadal expression, gon-2 is known to exhibit intestinal expression (Teramoto et al., 2005). To confirm this, we generated a strain expressing gon-2::GFP under the gon-2 promoter, which revealed expression in the gonads and intestine (Figure 7 – supplement 2). […] Together, these results indicate that gon-2 expression in the intestine and gtl-2 expression in the excretory cell is sufficient for *C. elegans* avoidance of *E. faecalis*.”

These mutants show the same anterior intestinal distention as wild-type animals on *E. faecalis* (New Figure 8 – Supplement 1). The revised manuscript reads: “Because gon-2(lf) and gtl-2(lf) animals display wild-type levels of anterior intestinal distention (Figure 8—figure supplement 1), we hypothesized that the effect observed in these animals was due to an inability to sense intestinal distention either directly or indirectly”.

Additionally, we have demonstrated that these genes play a role in pathogen learning by performing olfactory aversive learning assays with these mutants (New Figure 8C and D). The manuscript now reads:

“If these TRPM channels are somewhat involved in sensing anterior intestinal distention caused by *E. faecalis* infection, their loss should leave naïve olfactory preferences to E.

faecalis intact while diminishing negative associative learning. […] Altogether, these findings indicate that GON-2 and GTL-2 are required for the learned avoidance of *E. faecalis*, though their exact mechanistic role remains to be determined (Figure 8E).”

This olfactory aversive learning is almost certainly a form of negative associative learning as demonstrated by newly performed odor-pairing experiments with benzaldehyde and *E. faecalis* (New Figure 6C-E). The new relevant section of the manuscript reads:

“To strengthen the idea that avoidance of *E. faecalis* involves aversive learning, we paired an attractive odorant, benzaldehyde, with lawns of *E. faecalis* and exposed animals to this pairing for four hours (Figure 6C). […] Interestingly, animals with ablated AWB neurons displayed wild-type levels of naïve attraction but did not show a decrease in attraction upon training, indicating that AWB neurons may be required for this aversive learning process.”

We also find the question of how these channels are activated intriguing, but believe that addressing it is outside the scope of an already broad manuscript.

3. The role of npr-1 in pathogen avoidance is overstated, if not misconstrued. As it stands, the paper shows simply that, for npr-1(lf) animals, high O2 is a more potent aversive stimulus than is E.f. This is fine, but I don't see how it leads to the idea that npr-1 activity is downstream of pathogen exposure or detection, as is indicated in Figures5E and 7E, or that pathogen avoidance is "regulated" by npr-1, as stated in the first sentence of the Discussion.

This is a very good point and the manuscript has been re-written to reflect this:

“Furthermore, when animals were taken out of the low oxygen environment and allowed to roam at atmospheric oxygen levels for one-hour, wild-type animals remained off the *E. faecalis* lawns while both npr-1(lf) strains migrated back onto the lawns (Figure 4C), indicating that hyperoxia avoidance is a more potent aversive stimulus than *E. faecalis* in npr-1(lf) animals. […] This study establishes that *E. faecalis* infection in *C. elegans* leads to anterior intestinal distention which results in a rapid pathogen avoidance behavior, opposed by NPR-1-mediated hyperoxia avoidance.”

The figures have also been modified accordingly (New Figures 5E and 8E) and the title of this section edited to remove reference to NPR-1.

Reviewer #2 (Recommendations for the authors):This is a really excited set of findings and if the hypotheses are fully supported, I think this would result in a publication of very broad interest.The main concerns I have are related to the strength and specificity of the assertions made in each of the major finding areas I described. They are repeated below along with some suggestions that I think could strengthen these arguments.1. Anterior intestinal distension causes rapid behavioral avoidance.I found the authors' previous demonstrations linking distension to behavioral avoidance in Pseudomonas quite impressive and in my opinion the experiments using nol-6 to disrupt distension were particularly important in establishing that. In this case, because the behavioral avoidance is so rapid, I think demonstrating causality is even more challenging and important, if you intend to assert this as strongly as is currently written. Given its role in peptide processing, I'd suggest that solely using aex-5 mutations to establish sufficiency is not ideal here.I recognize that there aren't currently any tools available to block bloating via *E. faecalis* since nol-6 RNAi does not, but more carefully comparing the temporal progression of bloating relative to behavioral changes might be the next best approach. Lawn avoidance behavior is so rapid – is it really plausible that bloating might occur in minutes? Perhaps you could measure the intestine of animals that rapidly avoid compared to those that do not to see if there is a bias? Colonization was visualized at 4hr but is suggested to cause distension, which of these occurs first?Also, it might be that certain components of behavioral avoidance are driven by physical bloating and others are not. For example, ASE, AWC and AWB are all involved in lawn avoidance but ASE is not needed for olfactory choice.

We thank the reviewer for the comment about our previous work, and for making the connection to the present study. We agree that a demonstration of causality is lacking, and the title for the distention section has been updated to reflect this: “Avoidance of *E. faecalis* follows anterior intestinal distention and is independent of virulence”.

We unfortunately were not able to find a way to either cause bloating with *E. faecalis* infection or block bloating, and so we indeed did take the next best approach as suggested. Surprisingly, bloating does occur within minutes, as worms that left the *E. faecalis* lawn for the first time showed significant bloating compared to animals that were still inside the lawn (New Figure 2 – Supplement 1). It therefore seems that anterior distention precedes lawn avoidance. The manuscript now reads:

“Because we first measured the anterior intestinal diameter at 1 hour, it was not clear whether distention preceded avoidance or vice versa. […] It is therefore likely that fast accumulation of *E. faecalis* in the anterior intestine induces an aversive response to the pathogen, leading to avoidance.”

Additionally, to more strongly rule out immune signaling, would it not be more definitive to use fshr-1, pmk-1 or bar-1 mutants?

The intention of this section of the manuscript was not to rule out the role of immune signaling in avoidance per se, but to show that early exposure to *E. faecalis* induces distention but not immune pathways. Having said this, we agree with the suggestion that the manuscript would be strengthened by testing various immune pathway mutants to more completely rule out immune signaling. We thus tested *fshr-1, pmk-1,* and *bar-1* mutants as suggested and found that none of them affected avoidance behaviors (New Figure 3 – Supplement 1). This is consistent with the dispensability of *pmk-1* and *fshr-1* for general aversive behavior [Kaletsky et al. 2020; Lee and Mylonakis 2017; Melo and Ruvkun 2013]. The revised manuscript reads: “Furthermore, loss-of-function mutants for three key immune signaling genes, pmk-1, fshr-1, and bar-1, displayed wild-type levels of avoidance of *E. faecalis* (Figure 3—figure supplement 1).”

2. Avoidance occurs via an aversive learning process.The olfactory plasticity data shown are quite strong and convincing. My main concern is the specificity of the description of learning in this case. In my opinion, aversive learning is typically used as a shorthand to describe an associative learning paradigm. I assume the authors are not proposing some form of non-associative learning such as habituation or sensitization. For the olfactory learning process to be associative, there should be specificity to the response to odors that were present during training. In my opinion, a change in olfactory preference from *E. faecalis* to *E. coli* doesn't directly demonstrate this specificity.One way to address this would be to pair an AWB or AWC-sensed odor with the bacteria during training, then test if responses to this odor is specifically altered. If it turns out that this is not associative learning, this would still be quite interesting as it might indicate there are mechanisms to generally downregulate attraction to distension-causing bacteria – but this type of scenario should probably be referred to as some sort of state-dependent plasticity, as opposed to learning. If the effect is associative, can you rule out a satiety effect? Are the AWC/AWC dependent behaviors related to learning while the ASE effects are not? This concern might be solved by altering the text or interpretations.

This was a very helpful suggestion and we set out to test exactly what was described: pair an attractive odor, benzaldehyde, with *E. faecalis* during training and then conduct odor chemotaxis assays for benzaldehyde, comparing trained vs. naïve animal chemotaxis (New Figures 6C-E). Doing so revealed that wild-type animals do form a negative association between the odor and *E. faecalis* exposure, such that chemotaxis to the normally attractive benzaldehyde is reduced. Interestingly, while animals with AWC ablated neurons were not even able to establish the naïve preference, animals with AWB ablated neurons were, but they were not able to form the negative association. The manuscript reads:

“To strengthen the idea that avoidance of *E. faecalis* involves aversive learning, we paired an attractive odorant, benzaldehyde, with lawns of *E. faecalis* and exposed animals to this pairing for four hours (Figure 6C). […] Interestingly, animals with ablated AWB neurons displayed wild-type levels of naïve attraction but did not show a decrease in attraction upon training, indicating that AWB neurons may be required for this aversive learning process.”

3. Intestinal gon-2 and gtl-2 may sense intestinal bloating.In my opinion, this section represents potentially the most exciting finding but also the area in which the data are currently least strong. While you haven't directly stated that these channels are intestinal bloating sensors, this is strongly implied and/or suggested throughout the manuscript and impact statements.

We have taken the suggestion and toned down the language and raised other possibilities throughout the manuscript. For example the introduction now states:

“While no mechanoreceptors have been found to be functional in the *C. elegans* intestine, it does express transient receptor (TRP) channels which could act as mechanoreceptors (Xiao and Xu, 2011). However, TRP channels are known to play many other roles, including but not limited to taste, thermoregulation, ion homeostasis, and pacemaker activity (Holzer, 2011).”

The Discussion section now reads:

“However, TRPM channels display many distinct functions in various settings, including the absorption of calcium and magnesium ions by TRPM6 and TRPM7 in the human colon, TRPM7-mediated pacemaker activity of interstitial cells of Cajal, and colonic nociception by mouse TRPM8 (Holzer, 2011; Mueller-Tribbensee et al., 2015). Thus, it remains unclear how exactly GON-2 and GTL-2 function in pathogen avoidance.”

While gon-2 and gtl-2 are strongly expressed in the intestine, they are also expressed in other tissues, which might include neurons. To establish their site of action they should perform tissue specific rescue experiments.

We have performed tissue specific rescue and conducted avoidance assays, showing that rescue in intestine for *gon-2*, and excretory cell for *gtl-2* is sufficient for avoidance behaviors (New Figures 7C and D). This does not rule out a role for these genes in other cell types, though previously published reports have implicated these tissues in other processes (Kwan et al., 2008; Teramoto et al., 2005). The section describing these new findings has been written to reflect this possibility:

“In addition to gonadal expression, gon-2 is known to exhibit intestinal expression (Teramoto et al., 2005). […] Together, these results indicate that gon-2 expression in the intestine and gtl-2 expression in the excretory cell is sufficient for *C. elegans* avoidance of *E. faecalis*.”

Are the intestines of gon-2/gtl-2 mutants distended when grown on *E. faecalis* or is colonization affected?

We have included pictures and measurements showing that these mutants are indeed distended when grown on *E. faecalis* (New Figure 8 – Supplement 1). The revised manuscript reads: *“*Because gon-2(lf) and gtl-2(lf) animals display wild-type levels of anterior intestinal distention (Figure 8—figure supplement 1) …”

If these channels act in the intestine, short of directly demonstrating a sensory function, it seems reasonable that they could play many different roles, which should probably be acknowledged in the text.

See response above where we have highlighted other possibilities.

Reviewer #3 (Recommendations for the authors):The title of this paper emphasizes their discovery of TRPM channels in this learned pathogen avoidance response. I thought their characterization of the neural circuitry involved is equally important and could be emphasized instead in the title. To further characterize the TRPM channels, the authors could consider demonstrating that GON2 and GTL-2 function in the intestine (through intestinal expression in the mutant background), which would support their hypothesis nicely. Alternatively, the authors could further implicate these loci by rescuing the avoidance phenotype by expressing these genes under their own promoters in the mutant background.

As described in the Essential Revisions, we have driven expression for both genes using their own promoters and using an intestinal promoter for *gon-2* and an excretory cell promoter for *gtl-2*, and conducted avoidance assays (New Figure 8C and D). Doing so established that expression in these tissues was sufficient for rescuing the avoidance phenotype. We believe that the discovery of these TRPM channels being involved in learned pathogen avoidance is the most exciting part of the study, with the new experiments performed further strengthening this argument. We have thus kept the original title. The revised manuscript reads:

“In addition to gonadal expression, gon-2 is known to exhibit intestinal expression (Teramoto et al., 2005). […] Together, these results indicate that gon-2 expression in the intestine and gtl-2 expression in the excretory cell is sufficient for *C. elegans* avoidance of *E. faecalis*.”